# SPECTRAL GREEDY CORESETS FOR GRAPH NEURAL NETWORKS

## ABSTRACT

The ubiquity of large-scale graphs in node-classification tasks significantly hinders the real-world applications of Graph Neural Networks (GNNs). Node sampling, graph coarsening, and dataset condensation are effective strategies for enhancing data efficiency. However, owing to the interdependence of graph nodes, coreset selection, which selects subsets of the data examples, has not been successfully applied to speed up GNN training on large graphs, warranting special treatment. This paper studies graph coresets for GNNs and avoids the interdependence issue by selecting ego-graphs (i.e., neighborhood subgraphs around a node) based on their spectral embeddings. We decompose the coreset selection problem for GNNs into two phases, a coarse selection of widely spread ego graphs and a refined selection to diversify their topologies. We design a greedy algorithm that approximately optimizes both objectives. Our spectral greedy graph coreset (SGGC) scales to graphs with millions of nodes, obviates the need for model pre-training, and is applicable to low-homophily graphs. Extensive experiments on ten datasets demonstrate that SGGC outperforms other coreset methods by a wide margin, generalizes well across GNN architectures, and is much faster than graph condensation.

## 1 INTRODUCTION

*Graph neural networks* (GNNs) have found remarkable success in tackling a variety of graph-related tasks (Hamilton, 2020), e.g., node classification and link prediction. However, the prevalence of large-scale graphs in real-world contexts (e.g., social, information, and biological networks) poses significant computational issues for training GNNs, as graphs in these domains frequently scale to millions of nodes and edges. Training a single model is costly, and this increases when training multiple times, for instance, to validate design choices like architectures and hyperparameters (Elsken et al., 2019). To tackle the above issues, we adopt a natural *data-efficiency* approach — simplifying the given graph data appropriately, with the goal of saving training time. In particular, we ask the following question: *how can we appropriately simplify graphs while preserving the performance of GNNs*?

A simple yet effective solution to simplify a dataset is *coreset selection*, despite other methods such as graph sparsification, graph coarsening, and graph condensation reviewed in the related work Section 5. Typically, the coreset selection approach (Toneva et al., 2018; Paul et al., 2021) finds subsets of data examples that are important for training based on certain heuristic criteria. The generalization of coreset selection to graph node/edge classification problems is then to find the important "subsets" of the given graph, e.g., nodes, edges, and subgraphs. This challenge arises from graph nodes' interdependence and GNNs' non-linearities. We focus on node classification in this paper as it is among the important learning tasks on graphs and is still largely overlooked.

In this paper, we find a new approach to formulate graph coreset selection for GNNs. It avoids GNN's interdependent nodes and non-linear activation issues by selecting ego-graphs, i.e., the subgraph induced by the neighborhood around a center node, based on their node embeddings in the graph-spectral domain. Our ego-graph-based coreset is inspired by two observations. **(1)** We find that most GNNs applied to large graphs follow the nearest-neighbor message-passing update rule and have ego-graph-like receptive fields. Thus, by selecting the ego-graphs (which is equivalent to selecting their center nodes), we avoid the problem of finding subsets of nodes and edges independently, which typically leads to complex combinatorial optimization; see Section 2. **(2)** Moreover, we identify that when expressing the node embeddings in the graph-spectral domain, the non-linear spectral

embeddings of GNNs on ego-graphs are "smooth" functions of the center node, i.e., nearby nodes are likely to have similar spectral embeddings on their corresponding ego-graphs (Balcilar et al., 2021), which we will theoretically justify under certain assumptions in Section 3.

Using (1) and (2), we propose approximating the GNN's spectral embedding using a subset of ego-graphs. *To approximate the spectral embedding with fewer ego-graphs* (which one-to-one correspond to their center nodes), one should select center nodes that are far from each other since nearby ones are likely to have similar embeddings, thus, being less informative. We derive an upper bound on the coreset objective independent of the spectral embedding. This enables us to find the coreset of center nodes without evaluating the GNN's spectral embeddings on any ego-graph. With the coreset objective substituted by the upper-bound, we adopt the greedy iterative geodesic ascent (GIGA) approach (Campbell and Broderick, 2018; Vahidian et al., 2020) to obtain the coresets.

The above procedure of selecting distant ego-graphs is sufficient to approximate the whole graph's spectral embedding well. However, the selected center nodes do not necessarily approximate the node classification loss well, and the topological information of ego-graphs is not considered. To approximate the GNN training loss, we propose to refine the coreset selection by filtering out some of the selected ego-graphs whose topologies are not distinctive enough. Since the transformation from the spectral to the spatial domain is a linear operation depending on the graph topology, the approximated spatial embeddings of ego-graphs will differ more when they have more distinctive topologies. Hence, we exclude ego-graphs with non-distinctive spatial embeddings to enhance efficiency. This is solved by the submodular coreset algorithm (Iyer et al., 2021; Kothawade et al., 2022) using greedy submodular maximization (Mirzasoleiman et al., 2020).

As a result, we decompose the ego-graph selection into two stages: a coarse selection of widely spread ego-graphs that approximate the whole graph's spectral embedding (as detailed in Eq. (NAC)) and a refined selection to approximate the node classification loss with improved sample efficiency (as detailed in Eq. (LCC)). Specifically, the first stage (which is solved via GIGA) **extrapolates** the graph to find distant center nodes over the original graph, and the second stage (which is solved via submodular maximization) **exploits** the candidate center nodes and keeps the most informative ones based on their topologies. We call this two-stage algorithm *spectral greedy graph coresets (SGGC)*. Our SGGC compresses node attributes of selected ego-graphs using low-rank approximations, maintaining efficient storage without compromising GNN performance, as shown in Section 6.

SGGC scales to large graphs, needs no pre-training, and performs well on both high- and low-homophily graphs. SGGC surpasses other coreset methods in our experiments on ten graph datasets. We show that the combined algorithm is better than applying either algorithm (GIGA or submodular maximization) individually. Moreover, SGGC matches graph condensation's performance (Jin et al., 2021), but is significantly faster and better generalizes across GNN architectures.

## 2 PROBLEM: GRAPH CORESETS FOR GNNS

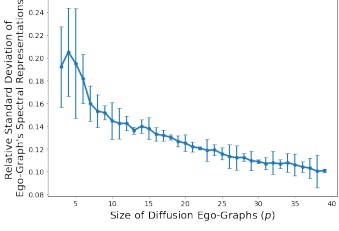
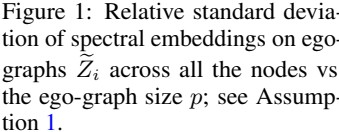
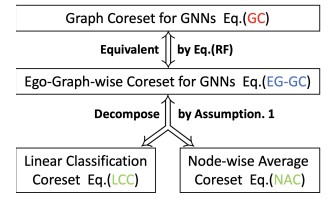
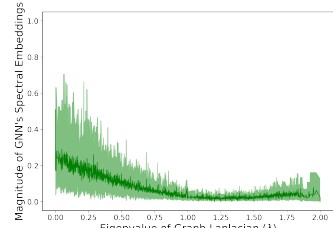

Figure 1: Relative standard deviation of spectral embeddings on ego-graphs $\widetilde{Z}_i$ across all the nodes vs. the ego-graph size $p$; see Assumption 1.

Figure 2: Conceptual diagram showing the theoretical analysis formulating the spectral greedy graph coresets (SGGC).

Figure 3: Spectral response of 2-layer GCNs on Cora. The spectral response corresponding to eigenvalue $\lambda_i$ is defined as $\|[U^\mathsf{T} f_\theta(A, X)]_{i,:}\| / \|[U^\mathsf{T} X]_{i,:}\|$.

We start by defining the downstream task for graph coresets and node classification with graph neural networks. For a matrix $M$, we denote its $(i, j)$-th entry, $i$-th row, $j$-th column by $M_{i,j}$, $M_{i,:}$, and $M_{:,j}$, respectively.

**Node classification on a graph** considers that we are given an (undirected) graph $G = (V = [n], E \subset [n] \times [n])$ with (symmetric) adjacency matrix $A \in \{0, 1\}^{n \times n}$, node features $X \in \mathbb{R}^{n \times d}$, node class labels $\mathbf{y} \in [K]^n$, and mutually disjoint node-splits $V_{\text{train}} \bigcup V_{\text{val}} \bigcup V_{\text{test}} = [n]$, where we assume the training split consists of the first $n_t$ nodes, i.e., $V_{\text{train}} = [n_t]$. Using a *graph neural network* (GNN) $f_\theta : \mathbb{R}_{\geq 0}^{n \times n} \times \mathbb{R}^{n \times d} \to \mathbb{R}^{n \times K}$, where $\theta \in \Theta$ denotes the parameters, we aim to find $\theta^* = \arg \min_\theta \mathcal{L}(\theta)$, where the training loss is $\mathcal{L}(\theta) := \frac{1}{n_t} \sum_{i \in [n_t]} \ell([f_{\theta,\lambda}(A, X)]_{i,:}, y_i)$. Here $Z_i := [f_{\theta,\lambda}(A, X)]_{i,:}$ is the output embedding for the $i$-th node. The node-wise loss function is $\ell(Z_i, y_i) := \text{CrossEntropy}(\text{softmax}(Z_i), y_i)$. The loss $\mathcal{L}(\theta)$ defined above is under the *transductive* setting, which can be generalized to the *inductive* setting by assuming only $\{A_{ij} \mid i, j \in [n_t]\}$ and $\{X_i \mid i \in [n_t]\}$ are used during training.

**Graph neural networks** (GNNs) can often be interpreted as iterative convolution over nodes (i.e., *message passing*) (Ding et al., 2021), where given inputs $X^{(0)} = X$,

$$X^{(l+1)} = \sigma(C_{\alpha^{(l)}}(A) X^{(l)} W^{(l)}) \quad \forall l \in [L], \tag{GNN}$$

and $X^{(L)} = f_\theta(A, X)$. Here, $C_{\alpha^{(l)}}(A)$ is the convolution matrix which is (possibly) parametrized by $\alpha^{(l)}$, $W^{(l)}$ is the learnable linear weights, and $\sigma(\cdot)$ denotes the non-linearity. See Appendix B for more details on GNNs.

**Graph coresets for GNNs** seek to select a subset of training nodes $V_w \subset V_{train} = [n_t]$ with size $|V_w| \leq c \ll n_t$ along with a corresponding set of sample weights such that the training loss $\mathcal{L}(\theta)$ is approximated for any $\theta \in \Theta$. Let $w \in \mathbb{R}_{\geq 0}^{n_t}$ be the vector of non-negative weights, we require the number of non-zero weights $\|w\|_0 := \sum_{i \in [n_t]} \mathbb{1}[w_i > 0] \leq c$, and hence the search space is $\mathcal{W} := \{w \in \mathbb{R}_{\geq 0}^{n_t} \mid \|w\|_0 \leq c\}$. The objective of graph coreset selection is

$$\min_{w \in \mathcal{W}} \max_{\theta \in \Theta} \Big| \sum_{i \in [n_t]} w_i \cdot \ell([f_\theta(A, X)]_i, y_i) - \mathcal{L}(\theta) \Big|, \tag{GC}$$

which minimizes the worst-case error for all $\theta$. However, the above formulation *provides nearly no data reduction* in terms of its size, since both the graph adjacency matrix $A$ and node feature matrix $X$ are still needed to compute the full-graph embeddings $f_\theta(A, X)$ in the coreset loss in Eq. (GC).

Since the goal of graph coresets is not only to reduce the number of labels, but more importantly, the data size, we should formalize how the subsets of $A$ and $X$ are selected in Eq. (GC) to make the objective practical. A natural convention, considered by related literature including graph condensation (Jin et al., 2021), is further *assuming* that only the node features of the selected nodes, $X_w = \{X_{i,:} \mid i \in V_w\}$, and the subgraph induced by the selected nodes, $A_w = \{A_{i,j} \mid i, j \in V_w\}$ are kept. Under this convention, the central problem of graph coresets changes to selecting the labeled nodes, as well as their node features and adjacencies, which we call as **node-wise graph coresets**,

$$\min_{w \in \mathcal{W}} \max_{\theta \in \Theta} \Big| \sum_{i \in [n_t]} w_i \cdot \ell([f_\theta(A_w, X_w)]_i, y_i) - \mathcal{L}(\theta) \Big|. \tag{N-GC}$$

However, since $A_w$ and $X_w$ are complex discrete functions of $w$, the above formulation leads to a complex combinatorial optimization, which we still need to learn how to solve. Moreover, posing $A_w$ and $X_w$ to be entirely determined by $w$ leads to sub-optimality. For example, it is likely that there exists another set of sample weights $w' \in \mathcal{W}$ such that using $A_{w'}$ and $X_{w'}$ in Eq. (N-GC) results in a smaller error.

**Alternative formulation of graph coresets.** The critical difference between Eq. (N-GC) and a typical coreset problem like Eq. (GC) is that the node-pair-wise relation encoded in the adjacency matrix $A$ forbids us to select nodes as independent samples. In this paper, we consider another formation of graph coresets, to avoid the non-independent issue. The idea is to use the property that most GNNs (especially those applied to large graphs) are "*local functions*" on the graph, i.e., the output embedding $Z_i = [f_{\theta,\lambda}(A, X)]_{i,:}$ of the $i$-th node may only depend on the close-by nodes $\{j \in [n] \mid d(i, j) < D\}$ where $d(i, j)$ denotes the shortest-path distance. Without loss of generality, we consider nearest-neighbor message passing (including GCN, GAT, and GIN), whose convolution/message-passing weight is non-zero $C_{i,j} \neq 0$ if and only if $i = j$ or $A_{i,j} = 1$. More specifically, we define the **receptive field** of a node $i$ for an $L$-layer GNN (Eq. (GNN)) as a set of nodes $V_i^L$ whose features $\{X_{j,:} \mid j \in V_i^L\}$ determines $Z_i$. For nearest-neighbor message passing,

it is easy to see $V_i^1 = \{i\} \cup \{j \in [n] \mid A_{i,j} = 1\} = \{j \in [n] \mid d(i,j) \leq 1\}$. Then by induction, for $L$-layer GNNs, the respective filed of node $i$ is $V_i^L = \{j \in [n] \mid d(i,j) \leq L\}$, which is exactly its depth-$L$ *ego-graph*. Here, we assume the GNN is $L$-layered, and the depth-$L$ **ego-graph** of node $i$, denoted by $G_i$, is defined as the induced subgraph of nodes within distance $L$. The above characterization of the "*local property*" of GNNs leads to the following equation,

$$\big[f_\theta(A, X)\big]_{i,:} = \big[f_\theta(A_{G_i}, X_{G_i})\big]_{1,:} \quad \forall i \in [n], \tag{RF}$$

where $A_{G_i}$ and $X_{G_i}$ denote the adjacencies and node features in the ego-graph $G_i$, respectively, where we always *re-number* the center node $i$ in $G_i$ as the first node.

**Ego-graph-wise graph coreset** can then be formulated by substituting Eq. (RF) into Eq. (GC),

$$\min_{w \in \mathcal{W}} \max_{\theta \in \Theta} \Big| \sum_{i \in [n_t]} w_i \cdot \ell\big([f_\theta(A_{G_i}, X_{G_i})]_{1,:}, \, y_i\big) - \mathcal{L}(\theta) \Big|. \tag{EG-GC}$$

Compared with node-wise graph coreset (Eq. (N-GC)), ego-graph-wise selection has the following advantages: (1) it avoids the non-independence issue as we are now selecting ego-graphs independently, i.e., whether $G_j$ ($j \neq i$) is selected will not affect the embedding $[f_\theta(A_{G_i}, X_{G_i})]_{1,:}$ of node $i$; (2) it is equivalent to the original objective (Eq. (GC)) which ensures optimality; and (3) although the adjacencies and node features in the union of selected ego-graphs $\bigcup_{i \in V_w} G_i$ are kept and their size could be $O(d_{\max}^L)$ times of the node-wise selected data (where $d_{\max}$ is the largest node degree), we find that we can highly compress the ego-graph node features via principal component analysis (PCA), depending on how far away the nodes are from the selected center nodes $V_w$, which eventually leads to comparable data size reduction. See Fig. 4 and Appendix A for details.

## 3 SPECTRAL GREEDY GRAPH CORESETS

Although selecting subsets of ego-graphs has many advantages, solving Eq. (EG-GC) is still *challenging* since the GNN $f_\theta$ is highly non-linear, and *expensive* since evaluating $A_{G_i}$ and $X_{G_i}$ requires finding the ego-graph $G_i$, which takes $O(d_{\max}^L)$ time. In this section, we propose an efficient yet effective approach to solve Eq. (EG-GC) that *avoids the non-linearities* in $f_\theta$ and does not require explicit evaluation of $A_{G_i}$ and $X_{G_i}$ for *any* node. The key idea is to re-write Eq. (EG-GC) in the *graph spectral domain*.

**Graph spectral domain** denotes the eigenspace of graph Laplacian (and the corresponding spectral feature space). Consider the symmetrically normalized Laplacian $L = I_n - D^{-1/2}AD^{-1/2}$ where $D$ is the diagonal degree matrix. Through eigendecomposition, $L = U\text{diag}(\lambda_1, \ldots, \lambda_n)U^\mathsf{T}$ where the eigenvalues $0 \leq \lambda_1 \leq \cdots \leq \lambda_n \leq 2$ and each column of $\mathbf{u}_i = U_{:,i}$ is an eigenvector. We can transform the features/embeddings to the spectral domain by left multiplying $U^\mathsf{T}$, e.g., $U^\mathsf{T}X$, where the $i$-th row $[U^\mathsf{T}X]_{i,:}$ is the features of eigenvalue $\lambda_i$.

Similarly, for each ego-graph $G_i$, we can find the *spectral representation of ego-graph embeddings*, denoted by $\widetilde{Z}_i = U_{G_i}^\mathsf{T} f_\theta(A_{G_i}, X_{G_i})$. To ease our analysis by making $\widetilde{Z}_i$ of different eigenvalues to have the same dimensions, we consider a slightly modified notion of ego-graphs, *diffusion ego-graphs*. Consider the diffusion matrix $P = \frac{1}{2}I_n + \frac{1}{2}D^{-1}A$, which is right stochastic (i.e., each row summing to 1) and describes a lazy-random walk on the graph. $P$ is simultaneously diagonalizable with $L$, whose eigenvalues are $1 \geq 1 - \frac{1}{2}\lambda_1 \geq \cdots \geq 1 - \frac{1}{2}\lambda_1 \geq 0$. We define the diffusion ego-graph $\widetilde{G}_i$ of node $i$ to be the induced subgraph of $\widetilde{V}_i^L = \{\text{indices of the } p \text{ largest entries of } [P^L]_{i,:}.\}$. For sufficiently large $p$, $\widetilde{G}_i \supseteq G_i$ for all $i$ and Eq. (RF) holds.

**Small variation of spectral embeddings on ego-graphs.** We start from a key observation that the variation of the spectral embeddings on ego-graphs $\widetilde{Z}_i = U_{G_i}^\mathsf{T} f_\theta(A_{G_i}, X_{G_i})$ across all the nodes $i \in [n]$ is small, when $p$ is not too small, for all possible $\theta$. This is formalized as follows.

**Assumption 1** (Bounded Variation of $\widetilde{Z}_i$). *For large enough graph size $n$ and ego-graph size $p$, we assume for any model parameter $\theta \in \Theta$, $RSD(\widetilde{Z}) := \sqrt{\frac{1}{n}\sum_{i \in [n]} \|\widetilde{Z}_i - \widetilde{Z}\|_F} / \|\widetilde{Z}\|_F < B$, i.e., the relative standard deviation (RSD) of $\widetilde{Z}$ is upper-bounded by a constant $B > 0$ independent of $\theta$, where $\widetilde{Z} = \frac{1}{n}\sum_{i \in [n]} \widetilde{Z}_i$ is the node-wise average of $\widetilde{Z}_i$.*

In Fig. 1, we plot $\mathrm{RSD}(\widetilde{Z}_i)$ versus $p$ on the Cora dataset (Yang et al., 2016), where we find the RSD drops vastly when $p$ increases. The intuition behind this phenomenon is that many real-world graphs (e.g., citation and social networks) are often *self-similar*, where the spectral representations of large-enough ego-graphs are close to the full graphs'.

**Approximately decompose the ego-graph-wise graph coreset objective Eq. (EG-GC) in spectral domain.** We can re-write Eq. (RF) in the spectral domain of each ego-graph as

$$\ell([f_\theta(A_{G_i}, X_{G_i})]_{1,:}, \ y_i) = \ell([U_{G_i}]_{1,:}\widetilde{Z}_i, y_i),\tag{SRF}$$

since $[f_\theta(A_{G_i}, X_{G_i})]_{1,:} = [U_{G_i}U_{G_i}^\mathsf{T} f_\theta(A_{G_i}, X_{G_i})]_{1,:} = [U_{G_i}]_{1,:}U_{G_i}^\mathsf{T} f_\theta(A_{G_i}, X_{G_i}) = [U_{G_i}]_{1,:}\widetilde{Z}_i$. We now denote $\mathbf{v}_i := [U_{G_i}]_{1,:}^\mathsf{T} \in \mathbb{R}^p$ (not an eigenvector), $\widetilde{\ell}_i(\widetilde{Z}) = \ell(\mathbf{v}_i^\mathsf{T}\widetilde{Z}, \ y_i)$, and $\widetilde{\mathcal{L}}(\widetilde{Z}) = \frac{1}{n_t}\sum_{i\in[n_t]}\widetilde{\ell}_i(\widetilde{Z})$. Since by Assumption 1, we assume $\widetilde{Z}_i \approx \widetilde{Z}$ for all $\theta \in \Theta$ and $i \in [n]$, we propose to approximately achieve the goal of ego-graph-wise coreset (Eq. (EG-GC)) by: (1) finding the subset of labeled nodes to approximate the average spectral embedding,

$$\min_{w^{\mathsf{a}}\in\mathcal{W}} \max_{\theta\in\Theta} \big\| \sum_{i\in[n_t]} w_i^{\mathsf{a}} \cdot \widetilde{Z}_i - \widetilde{Z} \big\|_F,\tag{NAC}$$

which we call **node-wise average coresets**; and (2) finding the subset of labeled nodes to approximate the node-classification loss,

$$\min_{w^{\mathsf{c}}\in\mathcal{W}} \max_{\widetilde{Z}} \big| \sum_{i\in[n_t]} w_i^{\mathsf{c}} \cdot \widetilde{\ell}_i(\widetilde{Z}) - \widetilde{\mathcal{L}}(\widetilde{Z})\big|,\tag{LCC}$$

where now the average spectral embedding $\widetilde{Z}$ is treated as an unknown parameter. Since $\widetilde{Z}$ is the output embedding and $\widetilde{\ell}_i(\widetilde{Z}) = \ell(\mathbf{v}_i^\mathsf{T}\widetilde{Z}, \ y_i)$ is a linear classification loss, Eq. (LCC) is the **linear classification coresets**. Although the optimal sample weights $w^{\mathsf{a}}$ and $w^{\mathsf{c}}$ (where the superscript $^{\mathsf{a}}$ stands for average and $^{\mathsf{c}}$ stands for classification) are different, we further require *the corresponding subsets of nodes coincide*, i.e., $V_{w^{\mathsf{a}}} = V_{w^{\mathsf{c}}}$, and this is realized by the combined coreset algorithm (see Algorithm 1). Moreover, given Assumption 1, if we can upper-bound the errors in Eqs. (NAC) and (LCC) through the combined coreset algorithm, we can show the approximation error on the node classification loss is also upper-bounded (see Theorem 1).

The remaining of this section analyzes how to solve the two coreset selection problems one by one, while we defer the combined greedy algorithm and theoretical guarantees to Section 4.

### 3.1 GRAPH NODE-WISE AVERAGE CORESETS

**Solving node-wise average coresets (Eq. (NAC)) approximately without evaluating the spectral embeddings.** For the node-wise average coresets (Eq. (NAC)), since the evaluation of a single spectral embedding $\widetilde{Z}_i$ is expensive, we ask: *is it possible to find the coresets approximately without evaluating any $\widetilde{Z}_i$?* Surprisingly, this is possible because the spectral embedding $\widetilde{Z}_i$ is a "*smooth*" function of nodes $i$ on the graph. Here, "*smothness*" refers to the phenomena that $\widetilde{Z}_i$ (as a function of node $i$) varies little across edges, i.e., $\widetilde{Z}_i \approx \widetilde{Z}_j$ if $A_{i,j} = 1$. The intuition behind this is simple: ego-graphs of connected nodes have a large overlap $\widetilde{G}_i \cap \widetilde{G}_j$, and thus the resulted output embedding is similar no matter what parameter $\theta$ is used.

**Spectral characterization of smoothness.** The spectral transformation can again be used to characterize the degree of smoothness since the eigenvalue $\lambda_i$ represents the smoothness of eigenvector $\mathbf{u}_i$. For an entry of the spectral embedding $[\widetilde{Z}_i]_{a,b}$, we can construct an $n$-dimensional vector $\widetilde{\mathbf{z}}^{(a,b)} = \big[[\widetilde{Z}_1]_{a,b}, \dots, [\widetilde{Z}_n]_{a,b}\big] \in \mathbb{R}^n$ by collecting the corresponding entry of the spectral embedding of each node. Then, we want to show the inner product $\langle\widetilde{\mathbf{z}}^{(a,b)}, \mathbf{u}_i\rangle$ is larger for smaller eigenvalue $\lambda_i$. Actually, this can be done by first considering the spectral representation of the inputs, i.e., $\widetilde{X}_i = U_{G_i}^\mathsf{T} X_{G_i}$, where we can similarly define $\widetilde{\mathbf{x}}^{(a,b)} = \big[[\widetilde{X}_1]_{a,b}, \dots, [\widetilde{X}_n]_{a,b}\big]$ and show that if the node features are *i.i.d.* unit Gaussians, then in expectation $\langle\widetilde{\mathbf{x}}^{(a,b)}, \mathbf{u}_i\rangle \propto (1 - \frac{1}{2}\lambda_i)^L$ (see Lemma 2 in Appendix A). Second, we note that the spectral behavior of message-passing GNNs $f_\theta(A, X)$ (Eq. (GNN)) is completely characterized by its convolution matrix (Balcilar et al., 2021) (see Fig. 3 for practical observations). Based on this, we can show the corresponding GNN function in the spectral domain $\widetilde{f}_\theta(\cdot) = U^\mathsf{T} f_\theta(A, U\cdot)$ is Lipschitz continuous if all of the linear weights $W^{(l)}$ in Eq. (GNN)

have bounded operator norms (see Lemma 3 in Appendix A). Based on these results, we can formally characterize the smoothness of spectral embeddings (see Proposition 4 in Appendix A).

**Upper-bound on the node-wise average error.** Following the work in (Linderman and Steinerberger, 2020; Vahidian et al., 2020), and based on Proposition 4, we can obtain an upper-bound on the node-wise average error $\|\sum_{i \in [n_t]} w_i^{\mathtt{a}} \cdot \widetilde{Z}_i - \widetilde{Z}\|_F \leq M \cdot \|P\mathbf{w}^{\mathtt{a}} - \frac{1}{n}\mathbb{1}\|$ (see Theorem 5 in Appendix A), where $\mathbf{w}^{\mathtt{a}} = \sum_{i \in [n_t]} w_i^{\mathtt{a}} \boldsymbol{\delta}_i \in \mathbb{R}^n$, $\boldsymbol{\delta}_i$ is the unit vector whose $i$-th entry is one, and $\mathbb{1}$ is the vector of ones. We then propose to optimize the upper-bound $\|P\mathbf{w}^{\mathtt{a}} - \frac{1}{n}\mathbb{1}\|$ which does not depend on $\widetilde{Z}_i$, enabling us to approximately solve the node-wise average coreset without evaluating a single $\widetilde{Z}_i$. (Vahidian et al., 2020) propose to optimize $\|P\mathbf{w}^{\mathtt{a}} - \frac{1}{n}\mathbb{1}\|$ using a variant of the greedy geodesic iterative ascent (GIGA) algorithm (Campbell and Broderick, 2018), and we follow their approach (see Section 4 for details).

## 3.2 SPECTRAL LINEAR CLASSIFICATION CORESETS

There are more available approaches to solve the linear classification coreset Eq. (LCC), and we adopt the submodular maximization formulation in (Mirzasoleiman et al., 2020).

**Submodular maximization formulation of linear classification coreset.** Following (Mirzasoleiman et al., 2020), we can show the approximation error in Eq. (LCC) can be upper-bounded by a set function $H(\cdot)$, i.e., $|\sum_{i \in [n_t]} w_i^{\mathtt{c}} \cdot \widetilde{\ell}_i(\widetilde{Z}) - \widetilde{\mathcal{L}}(\widetilde{Z})| \leq H(V_{w^{\mathtt{c}}})$, where $H(V_{w^{\mathtt{c}}}) := \sum_{i \in [n_t]} \min_{j \in V_{w^{\mathtt{c}}}} \max_{\widetilde{Z}} |\ell_i(\widetilde{Z}) - \ell_j(\widetilde{Z})|$ (see Lemma 6 in Appendix A). Then, by introducing an auxiliary node $\{i_0\}$, we can define a submodular function $F(V) := H(\{i_0\}) - H(V \cup \{i_0\})$, and formulate the coreset selection as a submodular set-cover problem. Due to the efficiency constraints, (Mirzasoleiman et al., 2020) propose to solve the submodular maximization problem instead, $\max_{w^{\mathtt{c}} \in \mathcal{W}} F(V_{w^{\mathtt{c}}})$, which is dual to the original submodular cover formulation. We follow this approach and adopt their CRAIG (CoResets for Accelerating Incremental Gradient descent) algorithm for the linear classification coreset. It is worth noting that, although the CRAIG formulation discussed above can be used to solve the original ego-graph-wise coreset problem (Eq. (EG-GC)) directly, it suffers from a much larger complexity as we have to forward- and backward-pass through the GNN all the time, and evaluate all ego-graph embeddings explicitly.

## 4 ALGORITHM AND THEORETICAL ANALYSIS

**The spectral greedy graph coresets (SGGC) algorithm.** We now describe how we combine the two greedy algorithms, GIGA and CRIAG, to achieve both objectives respectively, with an extra constraint that they find the same subset of nodes, i.e., $V_{w^{\mathtt{a}}} = V_{w^{\mathtt{c}}}$. The idea is to incorporate the submodular cost $F(V_{w^{\mathtt{c}}}) = F(V_{w^{\mathtt{a}}})$ into the SCGIGA's objective. Through the introduction of a hyperparameter $0 < \kappa < 1$, we change the objective of the node-wise average coreset to be $\|P\mathbf{w}^{\mathtt{a}} - \frac{1}{n}\mathbb{1}\| - \kappa F(V_{w^{\mathtt{a}}})$. Now, the submodular cost $F(V_{w^{\mathtt{a}}})$ can be understood as a selection cost, and the new objective can be solved by a relaxation on the GIGA algorithm, which is called SCGIGA as discussed in (Vahidian et al., 2020). The complete pseudo-code is shown below (see Appendix A for more details).

---
**Algorithm 1:** Spectral greedy graph coresets (SGGC).

**Input:** Diffusion matrix $P = \frac{1}{2}I_n + \frac{1}{2}D^{-1}A$, coreset size $c$, hyperparameter $0 < \kappa < 1$.

1   Initialize weights $w_0^{\mathtt{a}} \leftarrow \mathbf{0}, w_0^{\mathtt{c}} \leftarrow \mathbf{0}$

2   **for** $t = 0, \ldots, c - 1$ **do**

3      Compute $P(w_t^{\mathtt{a}}) = \sum_{i \in [n_t]} [w_t^{\mathtt{a}}]_i \frac{P_{:,i}}{\|P_{:,i}\|}$

4      Compute $\mathbf{a}_t \leftarrow \frac{1 - \langle 1, P(w_t^{\mathtt{a}}) \rangle P(w_t^{\mathtt{a}})}{\|1 - \langle 1, P(w_t^{\mathtt{a}}) \rangle P(w_t^{\mathtt{a}})\|}$, and for each $i \in [n_t]$, $\mathbf{b}_t^i \leftarrow \frac{P_i - \langle P_i, P(w_t^{\mathtt{a}}) \rangle P(w_t^{\mathtt{a}})}{\|P_i - \langle P_i, P(w_t^{\mathtt{a}}) \rangle P(w_t^{\mathtt{a}})\|}$

5      Find subset $V_t = \{i \in [n_t] \mid \langle \mathbf{a}_t, \mathbf{b}_t^i \rangle \geq \kappa \cdot \max_{j \in [n_t]} \langle \mathbf{a}_t, \mathbf{b}_t^j \rangle\}$

6      Select node $i^* = \arg\max_{i \in V_t} F(\{i\} \cup V_{w^{\mathtt{a}}}) - F(V_{w^{\mathtt{a}}})$

7      Compute $\zeta_0 = \langle \frac{1}{\sqrt{n}}, P_{i^*} \rangle, \zeta_1 = \langle \frac{1}{\sqrt{n}}, P(w_t) \rangle, \zeta_2 = \langle P_{i^*}, P(w_t) \rangle$, and $\eta_t \leftarrow \frac{\zeta_0 - \zeta_1 \zeta_2}{(\zeta_0 - \zeta_1 \zeta_2) + (\zeta_1 - \zeta_0 \zeta_2)}$

8      Update weights $w_{t+1}^{\mathtt{a}} \leftarrow \frac{(1 - \eta_t) w_t^{\mathtt{a}} + \eta_t \boldsymbol{\delta}_{i^*}}{\|(1 - \eta_t) P(w_t^{\mathtt{a}}) + \eta_t P_{i^*}\|}$

9   Compute $[w^{\mathtt{a}}]_i \leftarrow \frac{1}{n\|P_{:,i}\| \|\sum_{j \in [n_t]} [w_c^{\mathtt{a}}]_j P_{:,j}\|} [w_c^{\mathtt{a}}]_i \quad \forall i \in [n_t]$

10   Compute $w^{\mathtt{c}} = \sum_{j \in [n_t]} \mathbb{1}\{i = \arg\min_{k \in V_{w^{\mathtt{a}}}} \max_{\widetilde{Z}} |\widetilde{\ell}_j(\widetilde{Z}) - \widetilde{\ell}_k(\widetilde{Z})|\}$

11   Combine $w_i \leftarrow w_i^{\mathtt{a}} \cdot w_i^{\mathtt{c}}$ for each $i \in [n_t]$, and normalize $w \leftarrow w/\|w\|_1$

12   **return** *coreset $V_w$, weights $w$*

---

**Theoretical guarantees of SGGC.** Based on the correctness theorems of SCGIGA and CRAIG, and Assumption 1, we can prove the following error-bound on the node-classification loss, which shows SGGC approximately solves the graph coresets problem (Eq. (GC)) (see Appendix A).

**Theorem 1** (Error-Bound on Node Classification Loss)**.** *If both Eq. (N-GC) and Eq. (LCC) have bounded errors and Assumption 1 holds, then we have,* $\max_{\theta \in \Theta} \left| \sum_{i \in [n_t]} w_i^{\mathtt{a}} w_i^{\mathtt{c}} \cdot \ell\big([f_\theta(A_{G_i}, X_{G_i})]_{1,:}, \ y_i\big) - \mathcal{L}(\theta) \right| < \epsilon$, *where $\epsilon$ does not depend on the coreset size $c$ and the number of training nodes $n_t$.*

# 5 RELATED WORK

Table 1: SGGC is better than other model-agnostic/based coresets, graph coarsening, and comparable to graph condensation. We train 2-layer GCNs on the coreset/coarsened/condensed graphs and report the test accuracy. OOT and OOM refer to out-of-time/memory.

| Dataset | Ratio | Model-Agnostic Coresets | | | | Model-Based Coresets | | | | | Graph Reduction | Ours | Data Condense | Oracle |
|---|---|---|---|---|---|---|---|---|---|---|---|---|---|---|
| | | Uniform | Herding | K-Center | Forgetting | Cal | CRAIG | Glister | GraNd | GradMatch | Coarsening | SGGC | GCond | Full Graph |
| Cora | 15% | 67.7±4.5 | 66.1±1.2 | 64.3±4.8 | 65.4±3.1 | 71.6±1.0 | 68.4±4.4 | 65.6±5.6 | **71.9±1.7** | 72.0±1.3 | — | **72.9±0.6** | — | |
| | 25% | 71.8±4.2 | 69.9±1.0 | 72.6±2.5 | 72.6±3.5 | 75.3±1.5 | 74.4±1.7 | 74.3±2.4 | 74.4±1.5 | 74.7±2.3 | 31.2±0.2 | **78.6±1.0** | 79.8±1.3 | 81.2±0.4 |
| | 50% | 78.3±2.2 | 70.8±0.4 | 78.9±1.0 | 76.1±1.1 | **80.7±0.5** | 78.2±2.0 | 78.3±2.0 | **79.3±0.8** | 80.2±0.5 | 65.2±0.6 | **80.2±0.8** | 80.1±0.6 | |
| Citeseer | 15% | 53.6±7.9 | 46.1±1.6 | 47.5±6.3 | 51.5±4.9 | 53.2±2.0 | 55.4±6.7 | 54.0±5.0 | 57.0±3.9 | 58.8±3.9 | — | **63.7±3.1** | — | |
| | 25% | 61.7±3.2 | 54.9±3.9 | 61.6±4.0 | 55.3±5.5 | 56.1±2.8 | 59.5±4.3 | **62.0±5.5** | 64.4±1.5 | **66.0±1.5** | 52.2±0.4 | **67.2±2.4** | 70.5±1.2 | 70.6±0.9 |
| | 50% | 66.9±1.7 | 68.7±0.5 | 65.6±1.6 | 67.6±0.8 | 68.2±0.8 | 67.9±2.2 | 67.9±1.4 | 70.5±0.8 | **70.7±0.5** | 59.0±0.5 | 68.4±0.9 | 70.6±0.9 | |
| Pubmed | 15% | 65.7±4.5 | 61.9±1.0 | **69.0±4.8** | 65.4±4.9 | 71.7±0.8 | **73.2±4.1** | 65.6±5.5 | 62.0±1.0 | 65.3±4.5 | — | **72.5±1.5** | — | |
| | 25% | 71.1±1.8 | 65.9±0.4 | **73.3±2.6** | 69.0±2.5 | **74.7±1.7** | 71.0±3.3 | 71.5±3.2 | 70.6±2.3 | 71.1±1.4 | — | **75.8±1.6** | — | 79.3±0.6 |
| | 50% | 75.3±1.1 | 72.2±0.6 | **77.8±1.3** | 72.5±2.1 | **77.3±1.1** | 74.4±1.7 | 75.1±2.0 | 76.0±1.2 | 74.8±1.1 | — | **76.5±0.6** | — | |
| Flickr | 0.2% | 47.1±1.6 | 45.5±1.3 | 46.9±0.9 | 46.0±1.2 | OOT | OOT | **48.0±1.0** | **48.1±0.4** | **48.2±0.2** | 41.9±0.2 | **48.4±0.8** | 46.5±0.4 | |
| | 1.0% | 48.4±1.1 | 46.7±0.3 | 47.5±0.9 | 47.7±2.4 | OOT | OOT | **48.5±0.7** | **48.8±0.5** | **48.7±0.6** | 44.5±0.1 | **49.0±0.6** | 47.1±0.1 | 49.1±0.7 |
| | 2.0% | 47.0±1.1 | 45.5±0.6 | 46.9±0.7 | 46.6±1.6 | OOT | OOT | 47.4±0.9 | OOM | 48.5±0.6 | — | **64.4±0.4** | — | |
| ogbn-arxiv | 0.5% | 58.4±1.5 | 45.7±4.4 | 56.8±2.8 | 55.5±2.4 | OOT | OOT | 57.2±2.1 | OOM | 53.4±1.9 | 43.5±0.2 | **59.7±1.5** | 63.2±0.3 | |
| | 1.0% | 62.0±0.9 | 47.6±0.4 | 60.7±0.8 | 60.4±1.9 | OOT | OOT | 62.2±1.3 | OOM | 56.5±1.7 | 50.4±0.1 | **62.5±0.9** | 64.0±0.4 | 70.9±0.2 |
| | 2.0% | 64.7±0.5 | 56.5±0.5 | 62.4±0.9 | 62.8±2.4 | OOT | OOT | 64.2±1.1 | OOM | 58.6±1.0 | — | **64.4±0.4** | — | |
| ogbn-products | 0.05% | **46.8±1.2** | 31.9±0.5 | 35.9±1.9 | 32.9±4.8 | OOT | OOT | OOM | OOM | OOM | — | 46.3±4.1 | — | 75.6±0.2 |
| | 0.15% | **53.0±1.0** | 36.5±0.3 | 47.6±0.8 | 42.0±3.7 | OOT | OOT | OOM | OOM | OOM | — | 53.6±1.2 | — | |
| Reddit | 0.1% | 27.4±4.6 | 18.5±3.5 | 22.5±4.5 | 26.4±1.0 | OOT | OOT | OOM | OOM | 19.4±3.5 | — | **38.4±3.4** | — | 92.2±0.6 |
| | 0.2% | 40.7±7.2 | 17.0±4.0 | 20.0±3.1 | 39.7±3.5 | OOT | OOT | OOM | OOM | 18.3±3.0 | — | **48.6±4.6** | — | |

In this section, we review general coreset methods, graph coresets, and other graph reduction methods, as well as graph condensation that adapts dataset condensation to graph (see Appendix C).

Early coreset selection methods consider unsupervised learning problems, e.g., clustering. **Coreset selection** methods choose samples that are important for training based on certain heuristic criteria. They are usually **model-agnostic**; for example, *Herding* coreset (Welling, 2009) selects the closest samples to the cluster centers. *K-center* coreset (Farahani and Hekmatfar, 2009) picks multiple center points such that the largest distance between a data point and its nearest center is minimized. In recent years, more coreset methods consider the supervised learning setup and propose many **model-based** heuristic criteria, such as maximizing the diversity of selected samples in the gradient space (Aljundi et al., 2019), discovering cluster centers of model embedding (Sener and Savarese, 2018), and choosing samples with the largest negative implicit gradient (Borsos et al., 2020).

**Graph coreset selection** is a non-trivial generalization of the above-mentioned coreset methods given the interdependent nature of graph nodes. The very few off-the-shelf graph coreset algorithms are designed for graph clustering (Baker et al., 2020; Braverman et al., 2021) and are not optimal for the training of GNNs.

**Graph sparsification** (Batson et al., 2013; Satuluri et al., 2011) and **graph coarsening** (Loukas and Vandergheynst, 2018; Loukas, 2019; Huang et al., 2021; Cai et al., 2020) algorithms are usually designed to preserve specific graph properties like graph spectrum and graph clustering. Such objectives often need to be aligned with the optimization of downstream GNNs and are shown to be sub-optimal in preserving the information to train GNNs well (Jin et al., 2021).

**Graph condensation** (Jin et al., 2021) adopts the recent *dataset condensation* approach which *synthesizes* informative samples rather than selecting from given ones. Although graph condensation achieves the state-of-the-art performance for preserving GNNs' performance on the simplified graph, it suffers from two severe issues: **(1)** extremely long condensation training time; and **(2)** poor generalizability across GNN architectures. Subsequent work aims to apply a more efficient distribution-matching algorithm (Zhao and Bilen, 2021b; Wang et al., 2022) of dataset condensation to graph (Liu et al., 2022) or speed up gradient-matching graph condensation by reducing the number of gradient-matching-steps (Jin et al., 2022). While the efficiency issue of graph condensation is

| Dataset Ratio | Selection Strategy | Model-Agnostic | | Model-Based | | Ablation Baselines | | | Ours | Oracle |
|---|---|---|---|---|---|---|---|---|---|---|
| | | Uniform | K-Center | CRAIG | Glister | CRAIG-Linear | SCGIGA | **SGGC** | | Full Graph |
| Cora 25% | Node | 63.3±2.7 | 67.7±2.7 | 64.6±4.2 | 61.9±5.5 | 64.1±4.0 | 63.0±2.0 | 70.3±1.2 | | |
| | Std. Ego | 74.3±2.4 | 72.7±3.9 | 74.5±3.3 | 73.7±1.9 | 73.0±3.4 | 75.9±1.5 | 77.5±0.9 | | 81.2±0.4 |
| | Diff. Ego | 73.7±1.1 | 72.6±2.2 | 72.0±3.3 | 74.0±2.3 | 72.7±3.1 | 76.7±1.9 | 76.8±1.0 | | |
| Citeseer 25% | Node | 58.1±3.0 | 52.0±3.3 | 58.2±3.9 | 55.1±3.0 | 57.2±3.0 | 55.1±2.8 | 60.8±1.7 | | |
| | Std. Ego | 61.8±4.8 | 56.8±4.4 | 60.0±5.6 | 59.7±5.9 | 61.7±5.8 | 53.4±1.8 | 67.1±1.5 | | 70.6±0.9 |
| | Diff. Ego | 61.7±3.2 | 61.6±4.0 | 59.5±4.3 | 61.9±5.5 | 59.5±3.8 | 54.6±2.7 | 67.2±2.4 | | |

Table 2: Selecting diffusion ego-graphs largely outperforms node-wise selection and achieves comparable performance to selecting standard ego-graphs with much smaller ego-graph sizes.

| Dataset | Cora | | Citeseer | | Pubmed | |
|---|---|---|---|---|---|---|
| Ratio | 25% | 50% | 25% | 50% | 25% | 50% |
| Full Graph | 81.2±0.4 | | 70.6±0.9 | | 79.3±0.6 | |
| CRAIG-Linear | 72.7±2.9 | 78.1±1.1 | 59.5±3.8 | 66.6±1.9 | 71.6±3.8 | 75.4±2.3 |
| SCGIGA | 76.7±1.4 | 78.3±1.0 | 54.6±2.8 | 66.9±1.1 | 69.8±0.7 | 74.1±0.4 |
| **SGGC (Ours)** | **78.6±1.0** | **80.2±1.1** | **67.2±2.4** | **68.4±0.9** | **75.8±1.6** | **76.5±0.6** |

Table 3: The complete SGGC algorithm is better than the node-wise average coreset (SCGIGA) and the linear classification coreset (CRAIG-Linear) individually.

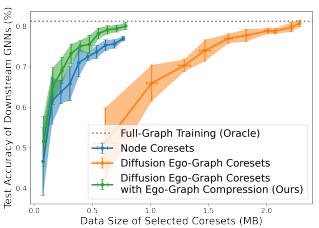

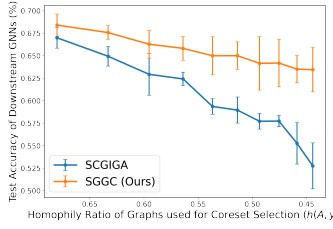

| Method | Architecture used during Compression | Downstream Architecture | | |
|---|---|---|---|---|
| | | GCN | SAGE | SGC |
| GCond | GCN | 70.6±3.7 | 60.2±1.9 | 68.7±5.4 |
| | SAGE | 77.0±0.7 | 76.1±0.7 | **77.7±1.8** |
| | SGC | **80.1±0.6** | 78.2±0.9 | 79.3±0.7 |
| **SGGC (Ours)** | N/A | **80.2±1.1** | **79.1±0.7** | **78.5±1.0** |

Figure 4: Test accuracy versus the selected data size of selecting nodes and diffusion ego-grpahs with/without PCA-based compression of node attributes.

Figure 5: SGGC is more robust than SCGIGA on low-homophily graphs. We select the coresets on the edge-added graph with lower homophily, but train and test GCNs on the original graph.

Table 4: SGGC generalizes better across GNN architectures than graph condensation (GCond) on Cora with a 50% ratio.

mitigated, the performance degradation on medium- and large-sized graphs (Jin et al., 2021) still renders graph condensation practically meaningless.

# 6    EXPERIMENTS

In this section, we demonstrate the effectiveness and advantages of SGGC, together with some important proof-of-concept experiments and ablation studies that verify our design. We also show the efficiency, architecture-generalizability, and robustness of SGGC. We define the coreset ratio as $c/n_t$[1], where $c$ is the size of coreset, and $n_t$ is the number of training nodes in the original graph. We train 2-layer GNNs with 256 hidden units and repeat every experiment 10 times. See Appendix D for implementation details and Appendix E for more results and ablation studies on more datasets.

**SGGC is better than other model-agnostic or model-based coresets and graph coarsening.**
Now, we demonstrate the effectiveness of SGGC in terms of the test performance (evaluated on the original graph) of GNNs trained on the coreset graph on seven node classification benchmarks with multiple coreset ratios $c/n_t$. Table 1 presents the full results, where SGGC consistently achieves better performance than the other coreset methods and the graph coarsening approach. Although graph condensation treats the condensed adjacency $A_w$ and node features $X_w$ as free learnable parameters (have less constraint than coreset methods), the performance is comparable to or even lower than SGGC on Cora and Flickr. The advantages of SGGC are often more significant when the coreset ratio is small (e.g., on Citeseer with a 15% ratio), indicating that SGGC is capable of extrapolating on the graph and finding informative ego-graphs when the budget is very limited.

Apart from the three small graphs (Cora, Citeseer, and Pubmed), we also consider two mid-scaled graphs (Flickr and ogbn-arxiv), a large-scale graph (ogbn-products) with more than two million nodes, and a much denser graph (Reddit) whose average node degree is around 50. In Table 1, we see that when scaling to larger and denser graphs, many model-based coreset methods, graph coarsening, and graph condensation are facing severe efficiency issues. SGGC can run on ogbn-product with a coreset ratio $c/n_t = 0.05\%$ within 43 minutes, while all the other model-based coresets (except Forgetting), graph coarsening, and graph condensation run out of time/memory.

---

[1]Some paper like Jin et al. (2021) defines this ratio as $c/n$, which could be small even if we keep all training/labeled nodes, i.e., $c = n_t$ (e.g., on Cora and Citeseer) and is often misleading.

**Selecting diffusion ego-graphs is better than selecting nodes and standard ego-graphs.** We also verify that selecting diffusion ego-graphs is advantageous to selecting nodes. In Fig. 4, we show that we can compress the diffusion ego-graphs to achieve data size comparable with node-wise selection without noticeably lowering the performance. Ego-graph compression is based on the principal component analysis (PCA), and we compress the node features more when they are far away from the selected center nodes (see Appendix A). In Table 2, we compare the performance of various coreset methods with the three selection strategies, including selecting the standard ego-graphs or diffusion ego-graphs. Not surprisingly, we see ego-graph selection strategies largely outperform node-wise selection (the largest gap is around 8%). Although selecting standard and diffusion ego-graphs often lead to similar performance, we note that, by selecting diffusion ego-graphs, we can achieve comparable performance with ego-graph-size $p = 8$ or 16 on most datasets, which is much smaller than the average size of standard ego-graphs for $L = 2$, e.g., around 36 on Cora.

**Ablation study: the two-stage SGGC algorithm is better than each stage individually.** It is important to verify that the combined coreset objective is better than the node-wise average coreset (Eq. (N-GC), implemented as SCGIGA (Vahidian et al., 2020) with zero selection cost) and the linear classification coreset (Eq. (LCC), implemented as the CRAIG algorithm (Mirzasoleiman et al., 2020) with a linear model, denoted by CRAIG-Linear) individually (see Appendix D for details). In Table 3, we see SGGC is consistently better than **(1)** CRAIG-Linear (outperformed by 3.8% on average), which over-simplifies GNNs to a linear classifier and completely ignores the graph adjacency and **(2)** SCGIGA (outperformed by 4.4% on average), which relies on a possibly wrong assumption that the node-wise classification loss is a "smooth" function of nodes over the graph. Moreover, we find SGGC is more robust than SCGIGA, against the variations of *homophily* in the underlying graph (as shown in Fig. 5), where the homophily is defined as $h(A, \mathbf{y}) = \frac{1}{|E|} \sum_{(i,j) \in E} \mathbb{1}\{y_j = y_k\}$ (Ma et al., 2021) (i.e., how likely the two end nodes of an edge are in the same class). SCGIGA's performance is greatly degraded on low-homophily graphs because it assumes the node-wise classification loss to be a "smooth" function of nodes over the graph. When we decrease the graph homophily by randomly adding edges to Cora, this assumption cannot be guaranteed. Our SGGC does not suffer from this issue because the spectral embedding of ego-graphs is always a "smooth" function over graph (see Section 3).

**SGGC generalizes better than graph condensation and is more efficient.** Finally, we compare SGGC with graph condensation (Jin et al., 2021) in terms of the generalizability across GNN architectures and running time. GCond is model-dependent and generalizes poorly across architectures. In Table 4, we see the performance of graph condensation heavily relies on the GNN architecture used during condensation, while our SGGC is model-agnostic and generalizes well to various types of GNNs. Although the best performance of graph condensation is comparable to SGGC in Table 4, if we do not tune the architecture for condensation, it is much lower on average. Specifically, when using SGC for condensation, the test performance of GCond is comparable to SGGC's. However, when using other architectures, including GCN and SAGE during condensation, the test accuracy of GCond drops for at least 2% in all settings. In terms of running time, apart from the fact that GCond cannot scale to large graphs like ogbn-product, it is much slower than SGGC. On ogbn-arxiv with the coreset ratio $c/n_t = 0.05\%$, graph condensation runs for 494s while SGGC only requires 133s.

## 7 CONCLUSIONS

This paper proposes spectral greedy graph coreset (SGGC), a coreset selection method on graph for graph neural networks (GNNs), and node classification. For the theoretical limitations, we note that the small variation assumption of spectral embeddings on ego graphs may not hold for non-message-passing GNNs and very dense graphs. For the practical limitations, we address the problem that although SGGC is practically very efficient, similar to most of the coreset algorithms, it has a $O(cn_t n)$ time complexity. This hinders us from applying a large coreset ratio on very large graphs, which consequently bottlenecks the downstream GNN performance on the coreset graph. Future work may consider more efficient setups, e.g., online coreset selection and training on graphs with hundreds of millions of nodes. Considering broader impacts, we view our work mainly as a methodological contribution, which paves the way for more resource-efficient graph representation learning. Our innovations can enable more scalable ways to do large-network analysis for social good. However, progress in graph learning might also trigger other hostile social network analyses, e.g., extracting fine-grained user interactions for social tracking.

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
