# A    PROOFS AND MORE THEORETICAL DISCUSSIONS

In this section, we present the proofs and the complete lemmas or theorems for the theoretical results mentioned in the main paper. We divide this section into three parts; each corresponds to a section/subsection of the paper. We also describe the PCA compression algorithm to the ego-graph's node features at last.

## A.1    PROOFS FOR SECTION 3.1

We start from the theoretical results in Section 3.1. For this subsection, our goal is to show that the spectral embeddings of ego-graphs, as a function of the center node, is a "smooth" function on the graph, in the sense that when transformed into the graph spectral domain, their high-frequency components is relatively small.

The above-mentioned "smoothness" characterization of ego-graph's spectral embeddings is presented as Proposition 4 in the main paper. In order to prove Proposition 4, as analyzed in Section 3, we first need to prove the spectral representation of ego-graph's input features is "smooth" (Lemma 2) under some assumptions, and then show the GNNs we considered (including GCN (Balcilar et al., 2021)) are Lipschitz continuous in the spectral domain (Lemma 3, as a corollary of the Theorem 4 in (Balcilar et al., 2021)).

**Lemma 2** (**Smoothness of the Spectral Representation of Ego-graph's Input Features**). *For an arbitrary entry ($a$-th row, $b$-th column) of the spectral representations of the ego-graphs' features, $\widetilde{X}_i = U_{G_i}^\top X_{G_i}$, we denote $\widetilde{\mathbf{x}}^{(a,b)} = \left[[\widetilde{X}_1]_{a,b}, \ldots, [\widetilde{X}_n]_{a,b}\right]$. If all node features, i.e., all entries of $X$ are i.i.d. unit Gaussians, then $\mathbb{E}[\langle \widetilde{\mathbf{x}}^{(a,b)}, \mathbf{u}_i \rangle] = (1 - \frac{1}{2}\lambda_i)^L$.*

*Proof.* For simplicity and without loss of generality, we consider there is only one feature per node, i.e., $X \in \mathbb{R}^n$ is a column vector. Thus $\widetilde{X}_i \in \mathbb{R}^n$ is also a column vector. We consider the $k$-th entry of the spectral representation.

We have $[U_{G_i}^\top X_{G_i}]_k = [U_{G_i}^\top X_{G_i}]_k = \sum_{j \in V_i^L} [U_{G_i}^\top]_{k,j} X_j = \sum_{j \in [n]} [U^\top]_{k,j} X_j [P^L]_{j,i} = [U^\top \mathrm{diag}(X_1, \ldots, X_n) U (I - \frac{1}{2}\Lambda)^L U^\top]_{k,i}$. Due to $\mathbb{E}[U^\top \mathrm{diag}(X_1, \ldots, X_n) U] = I$. Thus we have $\mathbb{E}[\widetilde{\mathbf{x}}^{(k)}] \approx [(I - \frac{1}{2}\Lambda)^L U^\top]_{k,:}$, and $\mathbb{E}[\langle \widetilde{\mathbf{x}}^{(k)}, \mathbf{u}_i \rangle] = (1 - \frac{1}{2}\lambda_i)^L$. $\qquad\square$

**Lemma 3** (**Lipschitzness of GCN in Spectral Domain**). *For an $L$-layer GCN (Kipf and Welling, 2016), if all linear weights $W^{(l)}$ have bounded operator norm, then the corresponding GNN function in the spectral domain $\widetilde{f}_\theta(\cdot) = U^\mathsf{T} f_\theta(A, U\cdot)$ is Lipschitz continuous.*

*Proof.* For simplicity and without loss of generality, we consider $L = 1$. Since $W^{(1)}$ has bounded operator norm, and the non-linear activation function $\sigma(\cdot)$ is Lipschitz continuous, we only need to show the convolution matrix in the spectral domain, i.e., $\widetilde{C} = U^\top C U$ has bounded operator norm. This directly follows from the Theorem 4 in (Balcilar et al., 2021), where they show $\|U^\top C U \mathbf{u}_i\| \approx [1 - \bar{d}/(\bar{d} + 1)\lambda_i]$. $\qquad\square$

Now we prove Proposition 4 as a corollary of Lemmas 2 and 3.

**Proposition 4** (Smoothness of Spectral Embeddings). *Assuming the node features $X$ are i.i.d. Gaussians, and the $L$-layer GNN (Eq. (GNN)) have operator-norm bounded linear weights $W^{(l)}$, then the spectral embedding $\widetilde{Z}_i$ is smooth on graph, in the sense that, $\langle \widetilde{\mathbf{z}}^{(a,b)}, \mathbf{u}_i \rangle \leq M \cdot (1 - \frac{1}{2}\lambda_i)^L$ for some constant $M$, where $\widetilde{\mathbf{z}}^{(a,b)} = \left[[\widetilde{Z}_1]_{a,b}, \ldots, [\widetilde{Z}_n]_{a,b}\right]$ and $\mathbf{u}_i$ is the eigenvector corresponding to eigenvalue $\lambda_i$.*

*Proof of Proposition 4:* Given all the assumptions of Lemmas 2 and 3, we have that for any entry, $\langle \widetilde{\mathbf{x}}^{(a,b)}, \mathbf{u}_i \rangle = (1 - \frac{1}{2}\lambda_i)^L$ in expectation, and $\|\widetilde{Z}_i - \widetilde{Z}_j\| = \|U^\mathsf{T} f_\theta(A, U\widetilde{X}_i) - U^\mathsf{T} f_\theta(A, U\widetilde{X}_j)\| \leq M \cdot \|\widetilde{X}_i - \widetilde{X}_j\|$ for $i, j \in [n]$. This directly leads to that $\langle \widetilde{\mathbf{z}}^{(a,b)}, \mathbf{u}_i \rangle \leq M \cdot \langle \widetilde{\mathbf{x}}^{(a,b)}, \mathbf{u}_i \rangle = M \cdot (1 - \frac{1}{2}\lambda_i)^L$.
$\square$

Finally, we show the upper bound on the error of approximating the node-wise average.

**Theorem 5** (**Upper-bound on the Error Approximating Node-wise Average**). *Under all assumptions of Proposition 4, we have* $\| \sum_{i \in [n_t]} w_i^{\mathsf{a}} \cdot \widetilde{Z}_i - \widetilde{Z} \|_F \leq M \cdot \| P \mathbf{w}^{\mathsf{a}} - \frac{1}{n} \mathbb{1} \|$ *for some constant* $M > 0$.

*Proof.* The proof mostly follows from the proof of Theorem 1 in (Linderman and Steinerberger, 2020). We begin with decomposing an arbitrary $(a, b)$-th entry of $\sum_{i \in [n_t]} w_i^{\mathsf{a}} \cdot \widetilde{Z}_i - \widetilde{Z}$ in the spectral domain. $\sum_{i \in [n_t]} w_i^{\mathsf{a}} \cdot \widetilde{\mathbf{z}}_i^{(a,b)} - \frac{1}{n} \sum_{j \in [n]} \widetilde{\mathbf{z}}_j^{(a,b)} = \sum_{j \in [n]} (\sum_{i \in [n_t]} w_i^{\mathsf{a}} \boldsymbol{\delta}_i - \frac{1}{n}) \widetilde{\mathbf{z}}_j^{(a,b)} = \sum_{k \in [n]} \langle \sum_{i \in [n_t]} w_i^{\mathsf{a}} \boldsymbol{\delta}_i - \frac{1}{n}, \mathbf{u}_k \rangle \langle \mathbf{u}_k, \mathbf{z}^{(a,b)} \rangle \leq M' \cdot \sum_{k \in [n]} \langle \sum_{i \in [n_t]} w_i^{\mathsf{a}} \boldsymbol{\delta}_i - \frac{1}{n}, \mathbf{u}_k \rangle (1 - \frac{1}{2} \lambda_k)^L$. Thus we have, $\left| \sum_{i \in [n_t]} w_i^{\mathsf{a}} \cdot \widetilde{\mathbf{z}}_i^{(a,b)} - \frac{1}{n} \sum_{j \in [n]} \widetilde{\mathbf{z}}_j^{(a,b)} \right|^2 \leq M'^2 \cdot \sum_{k \in [n]} \left| \langle \sum_{i \in [n_t]} w_i^{\mathsf{a}} \boldsymbol{\delta}_i - \frac{1}{n}, \mathbf{u}_k \rangle \right|^2 (1 - \frac{1}{2} \lambda_k)^{2L} = M'^2 \| \sum_{k \in [n]} (1 - \frac{1}{2} \lambda_k)^L \langle \sum_{i \in [n_t]} w_i^{\mathsf{a}} \boldsymbol{\delta}_i - \frac{1}{n}, \mathbf{u}_k \rangle \mathbf{u}_k \|^2 = M'^2 \cdot \| P^L (\sum_{i \in [n_t]} w_i^{\mathsf{a}} \boldsymbol{\delta}_i - \frac{1}{n}) \| = M'^2 \cdot \| P \mathbf{w}^{\mathsf{a}} - \frac{1}{n} \mathbb{1} \|^2$. Thus we conclude $\| \sum_{i \in [n_t]} w_i^{\mathsf{a}} \cdot \widetilde{Z}_i - \widetilde{Z} \|_F \leq M \cdot \| P \mathbf{w}^{\mathsf{a}} - \frac{1}{n} \mathbb{1} \|$ where $M = M' \cdot pd$. Here $p$ is the (diffusion) ego-graph size and $d$ is the number of features per node. $\square$

## A.2 PROOFS FOR SECTION 3.2

The major theoretical result in Section 3.2 is that the approximation error on the empirical loss of the (spectral) linear classification problem can be upper-bounded by a set function (i.e., a function which only depends on the selected set of nodes), which is formally stated as follows.

**Lemma 6** (**Upper-bound on the Error Approximating Empirical Loss**). *For the set function* $H(\cdot)$ *defined as* $H(V) := \frac{1}{n} \sum_{i \in [n_t]} \min_{j \in V} \max_{\widetilde{Z}} |\ell_i(\widetilde{Z}) - \ell_j(\widetilde{Z})|$, *we have* $\min_{w^{\mathsf{c}} \in \mathcal{W}} \max_{\widetilde{Z}} | \sum_{i \in [n_t]} w_i^{\mathsf{c}} \cdot \widetilde{\ell}_i(\widetilde{Z}) - \widetilde{\mathcal{L}}(\widetilde{Z}) | \leq H(V_{w^{\mathsf{c}}})$.

*Proof.* The proof follows from Section 3.1 of (Mirzasoleiman et al., 2020). Consider there is a mapping $\zeta : [n] \to V$ and let $w_i^{\mathsf{c}}$ be the number of nodes mapped to $i \in V$ divided by $n$, i.e., $w_i^{\mathsf{c}} = \frac{1}{n} |\{j \in [n] \mid \zeta j = i\}|$. Then, $\sum_{i \in [n_t]} w_i^{\mathsf{c}} \cdot \widetilde{\ell}_i(\widetilde{Z}) = \sum_{j \in [n]} \widetilde{\ell}_{\zeta(i)}(\widetilde{Z})$ and $| \sum_{i \in [n_t]} w_i^{\mathsf{c}} \cdot \widetilde{\ell}_i(\widetilde{Z}) - \widetilde{\mathcal{L}}(\widetilde{Z}) | \leq \frac{1}{n} \sum_{j \in [n]} |\widetilde{\ell}_{\zeta(i)}(\widetilde{Z}) - \widetilde{\ell}_i(\widetilde{Z})|$. From this we can readily derive, $\min_{w^{\mathsf{c}} \in \mathcal{W}} | \sum_{i \in [n_t]} w_i^{\mathsf{c}} \cdot \widetilde{\ell}_i(\widetilde{Z}) - \widetilde{\mathcal{L}}(\widetilde{Z}) | \leq \frac{1}{n} \sum_{i \in [n_t]} \min_{j \in V} |\ell_i(\widetilde{Z}) - \ell_j(\widetilde{Z})|$, and by taking $\max_{\widetilde{Z}}$ on both sides we conclude the proof. $\square$

## A.3 PROOFS FOR SECTION 4

The proof for the error-bound on the node-classification loss (Theorem 1) consists of two parts. First, we want to show that Algorithm 1 can achieve both objectives (Eqs. (N-GC) and (LCC)) which follows from the Theorem 2 of (Vahidian et al., 2020), Theorem 5, and Lemma 6. Second, we show under Assumption 1 we can bound the approximation error on the node classification loss.

*Proof of Theorem 1:* First, using Algorithm 1 and by the Theorem 2 of Vahidian et al. (2020), we have $\| P \mathbf{w}^{\mathsf{a}} - \frac{1}{n} \mathbb{1} \| \leq \sqrt{\frac{1}{n} - (\frac{\kappa}{n})^2} \cdot O((1 - \kappa^2 \epsilon^2)^{c/2})$ for some $\epsilon > 0$ and where $c = |V_w|$ is the size of selected coreset. By Theorem 5, we have the upper-bound on the objective of the node-wise average coreset (Eq. (N-GC)), $\| \sum_{i \in [n_t]} w_i^{\mathsf{a}} \cdot \widetilde{Z}_i - \widetilde{Z} \|_F < M^{\mathsf{a}}$ for some $M^{\mathsf{a}} > 0$. Then, by Theorem 1 of (Vahidian et al., 2020), we know $F(V_w) = H(\{i_0\}) - H(V_w \cup \{i_0\}) \geq O(\frac{1-\kappa}{\sqrt{n}})$. Thus by Lemma 6, we have an upper-bound on, $\max_{\widetilde{Z}} |\ell_i(\widetilde{Z}) - \ell_j(\widetilde{Z})| \leq M^{\mathsf{c}}$ for some $M^{\mathsf{c}} > 0$ for any $i \in V_w$ and $j \in [n]$. Note the $w^{\mathsf{a}}$ and $w^{\mathsf{c}}$ above is the output of Algorithm 1 and we have $V_{w^{\mathsf{a}}} = V_{w^{\mathsf{c}}} = V_w$.

Second, by repeatedly using the second bound above, we can get $| \sum_{i \in [n_t]} w_i^{\mathsf{a}} w_i^{\mathsf{c}} \cdot \widetilde{\ell}_i(\widetilde{Z}_i) - \sum_{j \in [n_t]} \sum_{k \in [n]} w_j^{\mathsf{a}} \widetilde{\ell}_k(\widetilde{Z}_j) | \leq M^{\mathsf{c}}$, where $\zeta : [n] \to V_w$ is the mapping described in Lemma 6. While, by the first bound above, Assumption 1, the Jensen-bound and the Lipschitzness of $\widetilde{\ell}_k$ (the Lipschitz coefficient is an absolute constant), we can derive, $| \sum_{i \in [n_t]} w_i^{\mathsf{a}} \widetilde{\ell}_k(\widetilde{Z}_i) - \widetilde{\ell}_k(\widetilde{Z}_k) | \leq$

$|\sum_{i \in [n_t]} w_i^{\mathtt{a}} \widetilde{\ell}_k(\widetilde{Z}_i) - \widetilde{\ell}_k(\widetilde{Z})| + |\widetilde{\ell}_k(\widetilde{Z}) - \widetilde{\ell}_k(\widetilde{Z}_k)| \le O(M^{\mathtt{a}} + B\|\widetilde{Z}\|)$. Combining the two inequalities, we conclude the proof. $\square$

### A.4 COMPRESSING EGO-GRAPH'S NODE FEATURES VIA PCA

In the main paper, we described the algorithm to find the center nodes $V_w$ of the ego-graphs. We then find the union of those ego-graphs, which consists of nodes $V_w^{(L)} = \bigcup_{i \in V_w} V_i^L$ where $V_i^L$ is the set of nodes in the ego-graph centered at node $i$.

We then propose to compress the ego-graph's node features, which consists of $|V_w^{(L)}|d = \xi cd \le c \times p \times d$ floating point numbers, to have a comparable size with the node features of the center nodes, which consists of only $c \times d$ floating point numbers. Here $c$ is the size of the coreset, $p$ is the diffusion ego-graph size, $d$ is the number of features per node, and $\xi = |V_w^{(L)}|/c \le p$ because there may be overlaps between the ego-graphs. In practice, the overlaps are often large, and we expect $1 < \xi \ll p$.

We first quantize all the nodes' features to half-precision floating-point numbers, and then the desired compress ratio for all the non-center nodes is $1/(\xi - 1)$. We then compress the features of each non-center node $j$ according to its shortest path distance to the nearest center node, denoted by $d_{\min}(j)$. Assuming an $L$-layer GNN and thus $L$-depth ego-graphs, for $1 \le l \le L$, we find the set of nodes $= \{j \in V_w^{(L)} \mid d_{\min}(j) = l\}$, and compress their features by the principal component analysis (PCA) algorithm (which finds a nearly optimal approximation of a singular value decomposition). If we keep the $q_l$ largest eigenvalues, we only need $(|V_w^{(L,l)}| + d + 1)q$ half-precision floating point numbers to store the features of $V_w^{(L,l)}$, we find $q_l$ by formula $q_l = \frac{cd}{|V_w^{(L,l)}|+d+1}(\frac{1}{2})^l$, which satisfies the targeted compress ratio $\sum_{l=1}^{l=L}(|V_w^{(L,l)}| + d + 1)q_l \le cd$.

## B MESSAGE-PASSING GNNS

In this section, we present more results and discussions regarding the common generalized graph convolution framework.

**Notations.** Consider a graph with $n$ nodes and $m$ edges. Connectivity is given by the adjacency matrix $A \in \{0,1\}^{n \times n}$ and features are defined on nodes by $X \in \mathbb{R}^{n \times d}$ with $d$ the length of feature vectors. Given a matrix $C$, let $C_{i,j}$, $C_{i,:}$, and $C_{:,j}$ denote its $(i,j)$-th entry, $i$-th row, $j$-th column, respectively. Besides, we simplify the notion of $\{1, ..., L\}$ to $[L]$. We use $\odot$ to denote the element-wise (Hadamard) product. $\|\cdot\|_p$ denotes the entry-wise $\ell^p$ norm of a vector and $\|\cdot\|_F$ denotes the Frobenius norm. We use $I_n \in \mathbb{R}^{n \times n}$ to denote the identity matrix, $\mathbb{1}$ to denote the vector whose entries are all ones, and $\delta_i$ to denote the unit vector in $\mathbb{R}^n$ whose $i$-th entry is 1. And $\|$ represents concatenation along the last axis. We use superscripts to refer to different copies of the same kind of variable. For example, $X^{(l)} \in \mathbb{R}^{n \times f_l}$ denotes node representations on layer $l$. A Graph Neural Network (GNN) layer takes the node representation of a previous layer $X^{(l)}$ as input and produces a new representation $X^{(l+1)}$, where $X = X^{(0)}$ is the input features.

**A common framework for generalized graph convolution.** GNNs are designed following different guiding principles, including neighborhood aggregation (GraphSAGE (Hamilton et al., 2017), PNA (Corso et al., 2020)), spatial convolution (GCN (Kipf and Welling, 2016)), spectral filtering (ChebNet (Defferrard et al., 2016), CayleyNet (Levie et al., 2018), ARMA (Bianchi et al., 2021)), self-attention (GAT (Veličković et al., 2018), Graph Transformers (YaronLipman, 2020; Rong et al., 2020; Zhang et al., 2020)), diffusion (GDC (Klicpera et al., 2019), DCNN (Atwood and Towsley, 2016)), Weisfeiler-Lehman (WL) alignment (GIN (Xu et al., 2018a), 3WL-GNNs (Morris et al., 2019; Maron et al., 2019)), or other graph algorithms ((Xu et al., 2020; Loukas, 2019)). Despite these differences, *nearly all GNNs can be interpreted as performing message passing on node features, followed by feature transformation and an activation function.* Now we rewrite this expression according to one pointed out by (Balcilar et al., 2021) in the form:

$$X^{(l+1)} = \sigma\left(\sum_r C^{(r)} X^{(l)} W^{(l,r)}\right), \tag{1}$$

where $C^{(r)} \in \mathbb{R}^{n \times n}$ denotes the $r$-th convolution matrix that defines the message passing operator, $r \in \mathbb{Z}_+$ denotes the index of convolution, and $\sigma(\cdot)$ denotes the non-linearity. $W^{(l,r)} \in \mathbb{R}^{f_l \times f_{l+1}}$ is the learnable linear weight matrix for the $l$-th layer and $r$-th filter.

Within this common framework, GNNs differ from each other by the choice of convolution matrices $C^{(r)}$, which can be either fixed or learnable. A learnable convolution matrix relies on the inputs and learnable parameters and can be different in each layer (thus denoted as $C^{(l,r)}$):

$$C_{i,j}^{(l,r)} = \underbrace{\mathfrak{C}_{i,j}^{(r)}}_{\text{fixed}} \cdot \underbrace{h_{\theta^{(l,r)}}^{(r)}(X_{i,:}^{(l)}, X_{j,:}^{(l)})}_{\text{learnable}}, \tag{2}$$

where $\mathfrak{C}^{(r)}$ denotes the fixed mask of the $r$-th learnable convolution, which may depend on the adjacency matrix $A$ and input edge features $E_{i,j}$. While $h^{(r)}(\cdot, \cdot) : \mathbb{R}^{f_l} \times \mathbb{R}^{f_l} \to \mathbb{R}$ can be any learnable model parametrized by $\theta^{(l,r)}$. We re-formulate some popular GNNs into this generalized graph convolution framework (see Table 5 for more details).

Table 5: Summary of GNNs re-formulated as generalized graph convolution.

| Model Name | Design Idea | Conv. Matrix Type | # of Conv. | Convolution Matrix |
|---|---|---|---|---|
| GCN[1] (Kipf and Welling, 2016) | Spatial Conv. | Fixed | 1 | $C = \widetilde{D}^{-1/2} \widetilde{A} \widetilde{D}^{-1/2}$ |
| SAGE-Mean[2] (Hamilton et al., 2017) | Message Passing | Fixed | 2 | $\begin{cases} C^{(1)} = I_n \\ C^{(2)} = D^{-1}A \end{cases}$ |
| GAT[3] (Veličković et al., 2018) | Self-Attention | Learnable | # of heads | $\begin{cases} \mathfrak{C}^{(r)} = A + I_n \ \text{and} \\ h_{\boldsymbol{a}^{(l,r)}}^{(r)}(X_{i,:}^{(l)}, X_{j,:}^{(l)}) = \exp\big(\text{LeakyReLU}( \\ \quad (X_{i,:}^{(l)}W^{(l,r)} \| X_{j,:}^{(l)}W^{(l,r)}) \cdot \boldsymbol{a}^{(l,r)})\big) \end{cases}$ |

[1] Where $\widetilde{A} = A + I_n$, $\widetilde{D} = D + I_n$.     [2] $C^{(2)}$ represents mean aggregator. Weight matrix in (Hamilton et al., 2017) is $W^{(l)} = W^{(l,1)} \| W^{(l,2)}$.
[3] Need row-wise normalization. $C_{i,j}^{(l,r)}$ is non-zero if and only if $A_{i,j} = 1$, thus GAT follows direct-neighbor aggregation.

Most GNNs can be interpreted as performing message passing on node features, followed by feature transformation and an activation function, which is known as the common "generalized graph convolution" framework.

**GNNs that cannot be defined as graph convolution.** Some GNNs, including Gated Graph Neural Networks (Li et al., 2015) and ARMA Spectral Convolution Networks (Bianchi et al., 2021) cannot be re-formulated into this common graph convolution framework because they rely on either Recurrent Neural Networks (RNNs) or some iterative processes, which are out of the paradigm of message passing.

## C    MORE RELATED WORK

**Dataset condensation** (or distillation) is first proposed in (Wang et al., 2018) as a learning-to-learn problem by formulating the network parameters as a function of synthetic data and learning them through the network parameters to minimize the training loss over the original data. However, the nested-loop optimization precludes it from scaling up to large-scale in-the-wild datasets. (Zhao et al., 2020) alleviate this issue by enforcing the gradients of the synthetic samples w.r.t. the network weights to approach those of the original data, which successfully alleviates the expensive unrolling of the computational graph. Based on the meta-learning formulation in (Wang et al., 2018), (Bohdal et al., 2020) and (Nguyen et al., 2020; 2021) propose to simplify the inner-loop optimization of a classification model by training with ridge regression which has a closed-form solution, while (Such et al., 2020) model the synthetic data using a generative network. To improve the data efficiency of synthetic samples in the gradient-matching algorithm, (Zhao and Bilen, 2021a) apply differentiable Siamese augmentation, and (Kim et al., 2022) introduce efficient synthetic-data parametrization. Recently, a new distribution-matching framework (Zhao and Bilen, 2021b) proposes to match the hidden features rather than the gradients for fast optimization but may suffer from performance degradation compared to gradient-matching (Zhao and Bilen, 2021b), where (Kim et al., 2022) provide some interpretation.

Recent benchmark (Guo et al., 2022) of **model-based coreset methods** on image classification indicates *Forgetting* and *GraNd* are among the best-performing ones but still evidently underperform

the dataset condensation approach (see Appendix C). *Forgetting* (Toneva et al., 2018) measures the forgetfulness of trained samples and drops those that are not easy to forget. *GraNd* (Paul et al., 2021) selects the training samples that contribute most to the training loss in the first few epochs.

**Graph sampling** methods (Chiang et al., 2019; Zeng et al., 2019) can be as simple as uniformly sampling a set of nodes and finding their induced subgraph, which is understood as a graph-counterpart of uniform sampling of *i.i.d.* samples. However, most of the present graph sampling algorithms (e.g., ClusterGCN (Chiang et al., 2019) and GraphSAINT (Zeng et al., 2019)) are designed for sampling multiple subgraphs (mini-batches), which form a cover of the original graph for training GNNs with memory constraint. Therefore, those graph mini-batch sampling algorithms are effective graph partitioning algorithms and not optimized to find just one representative subgraph.

## D    IMPLEMENTATION DETAILS

**Packages and Hardware specs.**    Generally, our project is developed upon the Pytorch framework, and we use Pytorch Geometric (https://pytorch-geometric.readthedocs.io/en/latest/) to acquire datasets Cora, Citeseer, Pubmed, and Flickr, and utilize Ogb (https://ogb.stanford.edu/) to get dataset Arxiv. The coreset methods on the graph are implemented based on Guo et al. (2022) (https://github.com/patrickzh/deepcore) and our downstream GNN structures, GCN, GraphSage, and SGC, are implemented based on Jin et al. (2021) (https://github.com/ChandlerBang/GCond). The experiments are conducted on hardware with Nvidia GeForce RTX 2080 Ti(11GB GPU).

**Dataset statistics.**    https://www.overleaf.com/project/631b0be4d8392e655f1a2cb7 We adopt five graph datasets, Cora, Citeseer, Pubmed, Flickr, and Arxiv in our experiments. Cora, Citeseer, and Pubmed are citation datasets whose node features represent the most common words in the text, and two nodes, each representing a paper, are connected if either one cites the other. Flickr is an image network where images connect if they share common properties, such as similar figures or buildings. While Arxiv is a directed citation network that contains all computer science papers in Arxiv indexed by MAG[2]. A directed edge from paper A to paper B establishes if A cites B. Here is detailed information on the datasets.

Table 6:  Statistics of the datasets.

|          | Nodes  | Edges   | Features | Classes | Train (%) | Validation (%) | Test (%) |
|----------|--------|---------|----------|---------|-----------|----------------|----------|
| Cora     | 2708   | 10556   | 1433     | 7       | 5.2       | 18.5           | 76.3     |
| Citeseer | 3327   | 9104    | 3703     | 6       | 3.6       | 15.0           | 81.4     |
| Pubmed   | 19717  | 88648   | 500      | 3       | 0.3       | 2.5            | 97.2     |
| Flickr   | 89250  | 899756  | 500      | 7       | 50.0      | 25.0           | 25.0     |
| Arxiv    | 169343 | 1166243 | 128      | 40      | 53.7      | 17.6           | 28.7     |

From Table 6, we could see that Cora, Citeseer, and Pubmed are network data with nodes of no more than 20,000 and edges of less than 100,000, which can be deemed as small datasets. While Flickr (which has 89,250 nodes and 899,756 edges) and Arxiv (which has 169,343 nodes and 1,166,243 directed edges) are much larger datasets, and more than 50% of the nodes are training nodes, so testing GNN accuracy for model-based coreset methods could be both time and space consuming.

**Hyper-parameter setups of SGGC.**    For SGGC, there are three hyper-parameters: slack parameter (denoted as $\kappa$), max budget (denoted as $s$), and diffusion ego-graph size. (1) Slack parameter, denoted by $\kappa$, means we could pick the nodes whose alignment score has at least $\kappa$ of the highest alignment score of $v*$ in Algorithm 1. The $\kappa$ shows the balance of our algorithm between SCGIGA and CraigLinear. When $\kappa$ is 1, we pick nodes in the same way as SCGIGA. When we select set $\kappa$ to be 0, this algorithm ignores geodesic alignment and becomes the CraigLinear algorithm. Therefore, $\kappa$ determines which algorithm in SCGIGA and CraigLinear SGGC is more similar. (2) Max budget, denoted by $s$, is the maximum number of nodes we could pick for each epoch in line 6 of Algorithm 1. This procedure could help accelerate the selection process for large graphs such as Flickr and Arxiv. We aim to find the largest max budget, which could remain an unnoticeable drop in accuracy compared to the case when the max budget is 1. (3) Diffusion ego graph size, denoted by $p$, is the ego-graph size

---

[2]https://www.microsoft.com/en-us/research/project/microsoft-academic-graph/

we fix for every coreset node in the ego-graph selecting procedure. Since the diffusion ego-graph size of one node is smaller than its two-degree standard ego-graph, the GNN test performance becomes better when the diffusion ego-graph is larger. However, we need to control the size of the diffusion ego-graph so that the induced subgraph has comparable memory with that of the node-wise coreset selection methods. Therefore, our ablation experiment in this part aims to find the smallest ego-graph size for each dataset where test performance does not increase anymore.

So, according to our analysis of hyper-parameters, we design our hyper-parameter selecting strategy and list the selected hyper-parameters below.

Table 7: SGGC hyper-parameter setups

|          | Slack Parameter ($\kappa$) | Max Budget ($s$) | Diffusion Ego-Graph Size ($p$) |
|----------|------|------|------|
| Cora     | 0.999 | 1  | 16 |
| Citeseer | 0.5   | 1  | 8  |
| Pubmed   | 0.1   | 1  | 16 |
| Flickr   | 0.5   | 10 | 8  |
| Arxiv    | 0.1   | 10 | 8  |
| Products | 0.1   | 8  | 2  |
| 'Reddit2 | 0.1   | 8  | 8  |

In the hyper-parameter tuning process, we first determine the diffusion ego-graph size for different datasets under the fixed max budget, the slack parameter $kappa$, and the fraction ratio since this hyper-parameter does not significantly influence the experiment result. Hence, we could pick the smallest ego-graph size with which the model reaches the highest GNN test accuracy. Taking Cora and Citeseer as an example, the test accuracy becomes around the best when the ego-graph size is 16 for Cora and 8 for Citeseer. After that, we fix the ego-graph size as we just selected and then choose the slack parameter $\kappa$ for each dataset according to GNN test accuracy. For instance, according to Table 14, the best performance achieves when $\kappa = 0.999$ for Cora and $\kappa = 0.5$ for Citeseer. Finally, we fix the selected diffusion ego-graph size and slack parameter $\kappa$ to find the best max budget $s$. This step is only for Flickr and Arxiv. We try to increase the max budget from 1, and the aim is to find the largest max budget which does not sacrifice noticeable accuracy to accelerate the SGGC algorithm as much as possible. Therefore, based on Table 15, we observe that the accuracy drops out of the highest performance error bar when the max budget is larger than 10 for Flickr and Arxiv, so we choose the max budget as 10 for both of them.

**Selection strategy setups** The GNN training varies according to different selection strategies (node-wise selection, standard ego-graph selection, and diffusion ego-graph selection). (1) When we choose node-wise selection, we directly induce a subgraph composed of the coreset nodes and train GNN inductively. (2) For the standard ego-graph strategy, we union the 2-hop ego-graphs of every selected coreset node and use this node and the labeled coreset nodes to train GNN transductively. (3) The diffusion ego-graph selection strategy is slightly different from the standard one, which controls the ego-graph size $p$. If the size of the 2-degree ego graph of one node exceeds $p$, then we randomly cut the ego-graph size to $p$. If not, we try to increase the ego-graph degree to add more nodes until the node has an ego-graph size $p$. Here we point out that not all ego-graphs could reach the size $p$ because the connection component the node is in has a size smaller than $p$.

**Hyper-parameter setups of other model-agnostic coreset methods.** The model-agnostic coreset methods, Uniform, Herding, kCenterGreedy, and Cal, are determined algorithms that directly select coresets based on node features, thus are hyper-parameter free. Besides, it is also worth noting that those coreset selection methods do not utilize the graph structure information, i.e., the graph adjacency matrix. For details, see Guo et al. (2022)

**Hyper-parameter setups of model-based coreset methods.** The model-based method takes advantage of the node's gradient/hidden features in GNN to select the coresets. Our experiments are conducted on a pre-trained 2-layer GCN model, specifically GCNConv (feature dim.,256)-50%dropout-GCNConv (256, class num.)-log softmax. This model is trained for five epochs. The optimizer is Adam, with a learning rate of 0.01 and weight decay 5e-4. Besides, we also offer other pre-trained models, such as SGC and SAGE, in our released code repository. For Craig, we choose the submodular function to be the Facility Location. For GradMatch, we take the regularization

parameter in orthogonal matching pursuit to be one, following the default setting of Guo et al. (2022). Forgetting, Craig, Glister, and GraNd are hyper-parameter-free except for the pretraining part.

**Downstream GNN architectures.** We implement GCN, SGC, and GraphSage as the downstream GNN architecture. GCN has two convolution layers and a 50% dropout layer between them. SGC and GraphSage are both 2-layer without dropout. Those three models do not have batch normalization. The optimizers for the three models are all Adam equipped with a learning rate of 0.01 and weight decay of 5e-4. The hidden dimension is 256 in all three models. For every coreset selection of each dataset, we train the GNN 10 times, and the GNN is trained for 600 epochs every time.

# E    MORE EXPERIMENTAL RESULTS.

## E.1    TRAINING TIME WITH SGGC

As a coreset selection method, SGGC can significantly reduce the training time and memory complexities of GNNs on large graphs. Specifically, when SGGC identifies a coreset of $c$ nodes for a graph with $n_t$ training nodes (usually $c \ll n_t$), training GNNs on the SGGC coreset graph exhibits linear time and memory complexity to $c$ rather than $n_t$. In other words, SGGC achieves sublinear training time and memory complexities (to the original graph size) simultaneously. In Table 8, we have recorded the actual training time and memory of GNNs on the SGGC coreset graph and the original graph under different combinations of datasets and coreset ratios ($c/n_t$). 2-Layer GCN with 256 hidden dimensions is trained for 200 epochs. Our experimental results verify the sublinear complexities of training GNNs using SGGC.

Table 8: GNN training time on SGGC coreset graphs and original graphs.

| Dataset | Ratio | Training Time on SGGC Coreset Graph | Training Time on Original Graph | Training Memory on SGGC Coreset Graph | Training Memory on Original Graph |
|---------|-------|-------------------------------------|----------------------------------|----------------------------------------|----------------------------------|
| Flickr | 0.2% | 9.9 s | 45.4 s | 2.8 MB | 147.5 MB |
| Flickr | 1.0% | 10.7 s | 45.4 s | 3.1 MB | 147.5 MB |
| ogbn-arxiv | 0.5% | 23.4 s | 68.1 s | 17.1 MB | 963.1 MB |
| ogbn-arxiv | 1.0% | 28.0 s | 68.1 s | 4.56 MB | 963.1 MB |

## E.2    SGGC WITH JKNETS AND GRAPH TRANSFORMERS

We argue that SGGC is also applicable to GNNs with a large or even non-local receptive field. We address this argument for each assumption, respectively, with experimental support.

We tested our SGGC with JKNets (Xu et al., 2018b) on ogbn-arxiv and Reddit. Many complex model-based coreset algorithms run out of time/out of memory on ogbn-arxiv and Reddit, like in Table 1. On ogbn-arxiv, we use a 4-layer GCN-based JK-MaxPool. The other hyperparameters are the same as in Table 1. The test accuracy of JKNets trained on the coreset graphs obtained by different algorithms is in Table 9. We see SGGC is still better than the other baselines.

Table 9: Performance results of applying SGGC with JKNets.

| Dataset | Ratio | Uniform | Herding | k-Center | Forgetting | SGGC (Ours) | Full Graph (Oracle) |
|---------|-------|---------|---------|----------|-----------|-------------|---------------------|
| ogbn-arxiv | 0.5% | $55.0 \pm 1.1$ | $35.7 \pm 0.9$ | $49.1 \pm 1.8$ | $51.5 \pm 1.9$ | $56.2 \pm 0.7$ | $72.2 \pm 0.3$ |
| ogbn-arxiv | 1.0% | $59.8 \pm 1.5$ | $40.2 \pm 2.2$ | $53.6 \pm 0.6$ | $59.5 \pm 1.6$ | $60.5 \pm 0.8$ | $72.2 \pm 0.3$ |
| Reddit | 0.1% | $20.3 \pm 6.5$ | $13.7 \pm 2.5$ | $25.8 \pm 1.6$ | $19.8 \pm 5.5$ | $40.4 \pm 1.9$ | $96.5 \pm 0.8$ |
| Reddit | 0.2% | $32.3 \pm 6.1$ | $14.3 \pm 1.3$ | $28.0 \pm 2.7$ | $22.5 \pm 2.1$ | $42.8 \pm 1.0$ | $96.5 \pm 0.8$ |

We also tested SGGC with Graph Transformers (Shi et al., 2020) on ogbn-arxiv. The other hyperparameters are the same as in Table 1. The test accuracy of Graph Transformers trained on the coreset graphs obtained by different algorithms is as follows. We see SGGC is still better than the other baselines in most cases.

Table 10: Performance results of applying SGGC with Graph Transformer.

| Dataset | Ratio | Uniform | Herding | k-Center | Forgetting | SGGC (Ours) | Full Graph (Oracle) |
|---------|-------|---------|---------|----------|------------|-------------|---------------------|
| ogbn-arxiv | 0.5% | $35.2 \pm 2.9$ | $35.2 \pm 0.6$ | $47.6 \pm 1.6$ | $52.2 \pm 2.0$ | $52.9 \pm 1.3$ | $72.1 \pm 0.4$ |
| ogbn-arxiv | 1.0% | $58.3 \pm 1.2$ | $38.9 \pm 2.5$ | $51.0 \pm 1.9$ | $55.9 \pm 2.7$ | $56.1 \pm 2.3$ | $72.1 \pm 0.4$ |

### E.3 SGGC ON LOW-HOMOPHILY GRAPHS AND GRAPHS REQUIRING LONG-RANGE REASONING

We conduct experiments on two real-world low-homophily graphs, Chameleon (homophily ratio $h = 0.23$) and Squirrel (homophily ratio $h = 0.22$) (Rozemberczki et al., 2021). The homophily ratio is defined as $h = \frac{1}{|E|} \sum_{(i,j) \in E} \Bbbk\{y_j = y_k\}$, where $E$ is the set of edges, $y_i$ is the label of node $i$, and $\Bbbk\{\cdot\}$ is the indicator function. Generally, the homophily ratio $h$ describes how likely the two end nodes of an edge are in the same class. Common node classification benchmarks are often high-homophily; for example, the homophily ratio of Cora is around $0.81$. The Chameleon and Squirrel graphs are relatively small, allowing us to execute most baseline methods on them successfully. As shown in Table 11, we find our SGGC nearly always shows the best performance, which indicates SGGC is robust to and suitable for low-homophily graphs.

Table 11: Performance comparison low-homophily graphs.

| Dataset | Ratio | Uniform | Herding | K-Center | CRAIG | Forgetting | Glister | GradMatch | GraNd | Cal | SGGC (ours) | Full Graph (Oracle) |
|---------|-------|---------|---------|----------|-------|------------|---------|-----------|-------|-----|-------------|---------------------|
| Chameleon | 12.5% | $49.0 \pm 2.8$ | $46.3 \pm 0.8$ | $44.7 \pm 1.1$ | $49.2 \pm 3.8$ | $46.9 \pm 2.3$ | $50.7 \pm 2.2$ | $43.2 \pm 1.6$ | $42.5 \pm 1.9$ | $49.5 \pm 1.0$ | $48.1 \pm 2.2$ | $58.7 \pm 1.6$ |
| Chameleon | 25.0% | $39.7 \pm 4.0$ | $32.9 \pm 1.8$ | $29.9 \pm 1.7$ | $38.2 \pm 4.1$ | $32.4 \pm 3.2$ | $38.8 \pm 3.0$ | $33.5 \pm 2.9$ | $37.3 \pm 2.9$ | $40.8 \pm 2.9$ | $40.1 \pm 2.3$ | $58.7 \pm 1.6$ |
| Chameleon | 50.0% | $50.4 \pm 3.8$ | $46.9 \pm 1.9$ | $45.3 \pm 1.2$ | $50.2 \pm 2.5$ | $47.1 \pm 1.5$ | $50.8 \pm 2.8$ | $43.6 \pm 1.1$ | $43.2 \pm 1.4$ | $48.9 \pm 2.1$ | $49.4 \pm 1.9$ | $58.7 \pm 1.6$ |
| Squirrel | 12.5% | $39.2 \pm 1.0$ | $38.5 \pm 0.9$ | $38.0 \pm 0.8$ | $38.6 \pm 1.3$ | $35.6 \pm 0.5$ | $38.4 \pm 1.3$ | $36.8 \pm 0.7$ | $36.5 \pm 0.8$ | $38.0 \pm 0.5$ | $38.7 \pm 1.3$ | $44.5 \pm 0.4$ |
| Squirrel | 25.0% | $33.3 \pm 1.5$ | $31.6 \pm 0.7$ | $31.7 \pm 0.8$ | $33.3 \pm 1.7$ | $31.6 \pm 0.8$ | $33.3 \pm 1.2$ | $31.0 \pm 0.6$ | $30.4 \pm 1.0$ | $32.7 \pm 0.5$ | $34.1 \pm 1.2$ | $44.5 \pm 0.4$ |
| Squirrel | 50.0% | $37.4 \pm 1.2$ | $36.9 \pm 1.2$ | $36.4 \pm 1.8$ | $37.9 \pm 1.8$ | $35.8 \pm 0.6$ | $37.8 \pm 1.0$ | $34.5 \pm 1.0$ | $35.2 \pm 0.7$ | $36.5 \pm 0.5$ | $38.2 \pm 1.1$ | $44.3 \pm 0.4$ |

We now include comparison results between SGGC and coreset baselines on the PascalVOC-SP dataset (Dwivedi et al., 2022). PascalVOC-SP is a node classification dataset that requires long-range interaction reasoning in GNNs to achieve strong performance in a given task. With 5.4 million nodes, PascalVOC-SP is larger than all datasets in Table 1, and many complex model-based coreset algorithms run out of time or memory. The performance metric for PascalVOC is macro F1, and our SGGC consistently shows the best performance, as shown in Table 12. This indicates that SGGC is also applicable to node classification tasks that require long-range information.

Table 12: Performance comparison on PascalVOC-SP, a graph dataset that requires long-range interaction reasoning.

| Dataset | Ratio | Uniform | Herding | k-Center | Forgetting | SGGC (Ours) | Full Graph (Oracle) |
|---------|-------|---------|---------|----------|------------|-------------|---------------------|
| PascalVOC-SP | 0.05% | $0.060 \pm 0.015$ | $0.040 \pm 0.006$ | $0.050 \pm 0.013$ | $0.044 \pm 0.009$ | $0.069 \pm 0.011$ | $0.263 \pm 0.006$ |
| PascalVOC-SP | 0.10% | $0.073 \pm 0.018$ | $0.051 \pm 0.004$ | $0.068 \pm 0.009$ | $0.062 \pm 0.016$ | $0.080 \pm 0.008$ | $0.263 \pm 0.006$ |

### E.4 ABLATION STUDIES

As the ablation studies, we first discuss how three key factors, max budget, diffusion ego-graph size, and slack parameter(denoted as $\kappa$), affect the GNN test accuracy of our SGGC algorithm. We perform experiments on GCN and display the SGGC performance under different hyper-parameters on Citeseer, Cora, Flickr, and Arxiv. Then we explore whether different selection strategies affect GNN test accuracy on large graphs, Flickr, and Arxiv.

**Ego-graph size.** Diffusion ego graph size is an important hyper-parameter to determine in coreset selection. Given the diffusion ego-graph size of one node is smaller than its two-degree standard ego-graph, when the diffusion ego-graph is larger, the GNN test performance becomes better. However, we need to control the size of the diffusion ego-graph so that the induced subgraph has a comparable memory with that of the node-wise coreset selection methods. Therefore, our ablation experiment in this part aims to find the smallest ego-graph size for each dataset where test performance does not increase anymore.

Table 13: GNN test performance with different diffusion ego-graph size on Flickr and Arxiv

| Diffusion Ego-Graph Size ($p$) | 4 | 8 | 12 | 16 | 20 | 24 | 28 |
|---|---|---|---|---|---|---|---|
| Cora | 80.4±0.7 | 80.1±0.5 | 79.9±0.8 | 80.5±0.7 | 80.6±0.9 | 80.0±0.6 | 80.1±0.7 |
| Citeseer | 67.2±1.1 | 67.7±0.6 | 67.9±0.8 | 67.2±0.5 | 67.2±1.1 | 67.2±1.1 | 66.5±1.2 |

The diffusion ego-graph size on Cora and Citeseer seems to have no clear influence on test accuracy if the ego-graph size is larger than a very small positive integer.

**Slack parameter.** In Algorithm 1, slack parameter (i.e. $\kappa$) means we could pick the nodes whose alignment score has at least $\kappa$ of the highest alignment score of $v*$. This $\kappa$ shows the balance of our algorithm between SCGIGA and CraigLinear. When $\kappa$ is 1, we pick nodes like pure SCGIGA. When we select set $\kappa$ to be 0, this algorithm ignores geodesic alignment and becomes the CraigLinear algorithm. Therefore, $\kappa$ determines which algorithm in SCGIGA and CraigLinear SGGC is more similar.

Table 14: GNN test performance with different kappa on Cora and Citeseer

| Slack Parameter ($\kappa$) | CraigLinear ($\kappa$=0) | 0.001 | 0.1 | 0.25 | 0.375 | 0.5 |
|---|---|---|---|---|---|---|
| Cora | 78.1±1.1 | 76.2±1.2 | 75.8±1.6 | 76.1±0.9 | 76.6±1.2 | 76.7±2.0 |
| Citeseer | 66.6±1.9 | 63.1±6.6 | 67.8±1.8 | 69.1±1.7 | 69.7±2.3 | 69.8±1.5 |

| Slack Parameter ($\kappa$) | 0.625 | 0.75 | 0.9 | 0.999 | 0.9999 | SCGIGA ($\kappa$=1) |
|---|---|---|---|---|---|---|
| Cora | 76.9±1.5 | 77.9±0.9 | 75.0±1.3 | 80.4±0.8 | 80.0±0.7 | 78.3±1.3 |
| Citeseer | 69.2±1.8 | 69.0±1.5 | 68.4±2.6 | 69.1±0.9 | 68.2±0.5 | 66.9±1.1 |

From Table 14, the test performance keeps comparable from around 0.001 to at least as high as 0.75 for Cora. When $\kappa$ approximates 1, test performance reaches its highest. However, for Citeseer, the accuracy is the highest at around 0.5. This result shows that $\kappa$ selection highly varies according to datasets. It is interesting that if we plot the $\kappa$-accuracy graph, the curve drops dramatically at 0 and 1. This remains not clear why the result would change that dramatically.

**Max budget.** Max budget is the number of nodes we could pick for each epoch in line 6 of Algorithm 1. This procedure could help accelerate the selection process for large graphs such as Flickr and Arxiv. We aim to find the largest max budget which could remain an unnoticeable drop in accuracy compared to the case when the max budget is 1.

Table 15: GNN test performance with different max budget on Flickr and Arxiv

| Max Budget ($s$) | 1 | 5 | 10 | 15 | 20 | 25 | 30 |
|---|---|---|---|---|---|---|---|
| Flickr | 49.5±0.2 | 49.3±0.4 | 49.3±0.3 | 49.0±0.5 | 49.1±0.4 | 48.7±0.4 | 48.4±0.8 |
| Arxiv | 60.9±1.5 | 61.4±1.4 | 59.8±1.3 | 56.9±2.0 | 57.7±3.2 | 54.8±3.4 | 50.2±5.6 |

From this table, we could see that increasing the max budget does not affect test accuracy much before the budget reach as large as 10, so we could take the budget to 10 for both Flickr and Arxiv.

**Standard ego graph sizes** Recall that in Section 6, our SGGC adopts the diffusion ego-graph selection strategy because it could dramatically save memory storage by cutting the ego-graph size of each node while keeping the GNN test accuracy from a noticeable decrease. To see it more clearly, we compare diffusion ego-graph size and standard average ego-graph size for all datasets.

### E.5 MORE RESULTS ON LARGE GRAPHS

As a good continuation of Table 1, we conduct main coreset selection methods on large graphs, Flickr and Arxiv, to further understand the power of our SGGC.

Table 16: Ego-graph size for different selection strategy on datasets.

|  | Cora | Citeseer | Pubmed | Flickr | Arxiv |
|---|---|---|---|---|---|
| Diffusion ego-graph size | 16 | 8 | 16 | 8 | 8 |
| Standard ego-graph size | 36.8 | 15.1 | 60.1 | 875.7 | 3485.2 |

Table 17: GNN test performance on large graph with different fraction ratio

|  |  | Uniform | Herding | kCenterGreedy | Forgetting | Glister | GradMatch | SGGC |
|---|---|---|---|---|---|---|---|---|
| Flickr | 0.2% | 45.9±2.0 | 44.5±0.9 | 46.6±1.8 | 45.7±2.1 | 47.4±0.8 | 48.3±0.5 | 46.9±1.5 |
| | 1% | 47.4±1.8 | 46.3±0.5 | 46.7±0.9 | 47.4±1.9 | 48.4±1.0 | 48.5±0.6 | 48.6±0.7 |
| Arxiv | 0.1% | 47.2±6.2 | 33.8±1.3 | 41.5±5.8 | 43.6±6.1 | 41.5±8.6 | 45.6±3.9 | 41.1±6.8 |
| | 0.5% | 57.4±2.6 | 45.6±0.4 | 57.4±1.0 | 56.6±2.7 | 56.1±3.0 | 51.2±4.2 | 60.0±1.9 |

From Table 17, we could see that the performance is the highest on Flickr under a fraction rate 1% and on Arxiv under a fraction rate 0.5%. It is worth noting that in a large graph, when the fraction rate increases, the GNN accuracy also increases, which is different from what we have observed in small graphs such as Pubmed, Cora, and Citeseer.