# OpenReview forum: "Spectral Greedy Coresets for Graph Neural Networks"
_ICLR.cc/2024/Conference — Submitted to ICLR 2024_

### Official Review · Reviewer_EvwQ · 2023-10-28

**Soundness:** 3 good
**Presentation:** 3 good
**Contribution:** 3 good
**Rating:** 5
**Confidence:** 3

**Summary:**

This paper aims to apply the coreset method, for efficiently processing large data sets, to node classification by considering spectral embeddings of (diffusion ego nets). As the authors note, ego nets are similar to the receptive field used in standard Message Passing GNNs. And they also claim that spectral embeddings of ego nets are a smooth node feature. (They prove this in the case where the node features are iid Gaussians, which is NOT convincing)

**Strengths:**

Demonstrate model is scalable to large graphs and applicable to heterophilous datasets

**Weaknesses:**

It is a bit unclear to me why the propsed method is needed. There are already many other ways to scale GNNs to large networks as discussed here https://blog.twitter.com/engineering/en_us/topics/insights/2021/simple-scalable-graph-neural-networks

It is not clear to me how the sparsity constraint on the weights is enforced. I assumed you would use e.g. ell^1 regularization, but this doesn't seem to be done anywhere

The assumption for the theory, that the features are i.i.d. Gaussians, completely undercuts the utility of the results. The main utilitity of MPNNs  is that the there are informative node features which should be smooth along the graph.


Minor Issues

``We focus on node classification in this paper as it is among the important learning tasks on graphs and is still largely overlooked" - I think you should add the phrase "in the context of graph coresets" here.

**Questions:**

How do the numerical results compare to methods that do not do any size reduction. Obviously, this is an apples to oranges comparison, but it would be good to know how much is "lost" by restricting to coresets.

---

### Official Review · Reviewer_y3Xk · 2023-10-30

**Soundness:** 3 good
**Presentation:** 3 good
**Contribution:** 2 fair
**Rating:** 6
**Confidence:** 4

**Summary:**

This paper contributes with a novel approach to select training nodes in a very large graph, where
the processing of all of them via GNNs leads to OOM (Out-of-memory).  This problem is known as
"coreset selection" in the literature. The typical setting is to learn non-negative weights leading to
minimize the deviation between the "as per node loss" and the training loss (empirical risk): zero weight
means that the training node is not informative for learning. This is consistent with the "locality principle"
of GNNs where node embeddings mostly rely on those of their neighbors. The authors leverage this principle
by building quasi-independent node samples: aking subgraphs around training nodes and considering only
the features of the nodes in each subgraph. In particular, they propose to limit the subgraphs to ego-graphs (close
nodes, e.g. shortest path).

The main contribution of the paper is as follows. The authors formulate the coreset selection problems in terms of finding
local embeddings close to real ones (under the assumption that the error of this approximation is bounded). This is explained
via the spectral domain of the embedding, and this leads in turn to minimize an upper bound which is independent of the spectral
embeddings themselves: it depends on the diffusion (transition) matrix P. The minimization is done by a variant of the greedy geodesic iterative ascent (GIGA) algorithm.

The proposed algorithm is proven to outperform with other coreset selection algorithms, but in mid-sized such as Flicker. It is also tested
with very large graph where its greedy nature provides efficiency wrt the alternatives.

**Strengths:**

* Interesting methodology with spectral roots.
* Greedy enough for dealing with mid-large-very large graphs.

**Weaknesses:**

* Experimental results satisfy only the claim that this method is scalable (large and very-large graph). However, the fact that it does not outperfom the alternatives in Flickr (mid-size) suggests that either the assumption does not apply to these graphs or the ego-coresets are not that effective, or the upper bound is not very tight. Basically, this bound relates the diffusion matrix with the uniform distribution (as in the case of a mixing time in random walk).

**Questions:**

* Why Flickr, which is large enough has similar performance to the alternatives?
* To what extent the upper bound is informative in large graphs? Could be interesting to test powers of P (as in mixing times).
* To what extent the assumption  (and the ego-net representation) are valid in graphs like Flickr.

---

### Official Review · Reviewer_Zq6N · 2023-10-31

**Soundness:** 3 good
**Presentation:** 2 fair
**Contribution:** 3 good
**Rating:** 5
**Confidence:** 3

**Summary:**

This paper proposes a coreset selection procedure for GNN training (focusing on node classification tasks). In order to avoid node-dependence issues, as is typical, the authors look at ego-nets and pick the entire neighborhood at once. They then transform the search problem into the spectral space, which makes it easier to use submodular optimization (and greedy algorithm) and avoid redundancy. The intuition is that far-away nodes are less redundant (hence prior work based on k-means and k-center for coreset construction). Unlike previous work on graph condensation, this work looks at multiple hops in the ego-net, which reduces the "complexity" of the optimization problem by introducing a smoothness condition. This relies on the typical assumption that GNN essentially computes a "local" function. The multi-hop ego-net, though, adds to the size of the data, which the authors claim can be reduced by PCA.

**Strengths:**

- Timely contribution as data efficiency is becoming more and more important.
- Relatively efficient method, and high performance especially in the aggressive sampling regime.
- More efficient than graph condensation (close performance competition) and less sensitive to model architecture.

**Weaknesses:**

- Graph condensation sometimes performs better although it is less efficient and more sensitive to model architecture. Unlike what's claimed
at the end of Section 5, I do not observe degradation based on model size. If graph condensation could run on larger datasets, I'd expect close competition in terms of performance.
- Focuses on node classification. What can be done for other GNN tasks?
- Smoothness assumption is restrictive, at least in theory. How does it depend on L? Did you run a study on this?
- Time complexity is not very good, O(c*n_t*n), which is why for the larger datasets, the algorithm run on very small fraction of data. This is usually fine because with larger data, one often needs a small fraction to obtain fair performance. Here, though, the dent in performance for obgn-products and reddit is significant.

**Questions:**

See above

- Could you compare runtime complexity with the other methods? Why are model-based coresets so resource intensive?
- How do the selected examples differ for various methods? Can you get insights into what the right selection criterion is? Is the distance in the graph really important, or distance in the gradient space, etc.?
- Do you perform the selection adaptively? There are some works such as [https://proceedings.mlr.press/v216/abu-el-haija23a.html] which look at the curriculum learning approach for sampling. This one is for subgraph sampling. In fact, would there be any advantage in not selecting the entire ego-net of depth L, and rather sampling the nodes out selectively?

---

### Official Review · Reviewer_iC6X · 2023-11-01

**Soundness:** 2 fair
**Presentation:** 2 fair
**Contribution:** 2 fair
**Rating:** 3
**Confidence:** 5

**Summary:**

This work introduces a coreset (subgraph) selection method (SGGC) for graph neural networks (GNNs) to substantially accelerate the training process. The key idea is to identify the most relevant subgraph (e.g. the depth-L ego-graph) by leveraging the spectral embedding vectors of neighboring nodes. A theoretical analysis of the error bound for node classification loss has been provided.

**Strengths:**

1. Coreset selection is potentially useful for significantly accelerating the training of GNNs
2. The proposed coreset selection method based on ego-graph identification using spectral embedding is novel.

**Weaknesses:**

1. It is not clear if Assumption 1 is valid for typical GNN models. According to the theoretical analysis in Theorem 1 (that directly depends on Assumption 1), the error bound on node classification loss does not depend on the coreset size and the number of training nodes. However, the experimental results in Table 1 on large data sets show a significant accuracy drop due to the coreset selection process: 70.9 => 64.4 (ogbn-arxiv), 75.6 => 53.6 (ogbn-products), and 92.2 => 48.6 (Reddit).

2. The experimental results show this method always performs poorly on large graphs. It is necessary to explain the potential performance loss due to the coreset selection method. The accuracy drop may be due to the fact that the proposed ego-graph selection algorithm will not significantly alter the mid-range eigenvalues/eigenvectors, but can substantially affect the longe-range (first few) Laplacian eigenvalues/eigenvectors which encode the global (structural) graph properties.

3. More trade-off analysis (using different ratios) should be conducted for large data sets to allow a better understanding of the accuracy drop due to coreset selection.

**Questions:**

1. What are the most important (sensitive) hyperparameters for the proposed coreset algorithm?

---

### Official Review · Reviewer_Yh7x · 2023-11-01

**Soundness:** 2 fair
**Presentation:** 3 good
**Contribution:** 2 fair
**Rating:** 3
**Confidence:** 4

**Summary:**

Motivated by the scalability challenge of training large GNN models, this paper studies the problem coreset selection for GNNs. More concretely, the authors define the coreset for GNN as a subset of labeled nodes, and the goal is to select a small coreset so that if we train the model with the loss computed only on the coreset, the testing accuracy should be similar to training on the full labeled set. The authors cast this as an optimization problem. The problem is intractable so the author simplifies the problem significantly according to some observations and assumptions. Then using existing techniques to obtain the coreset. In the experiments, the proposed coreset framework exhibits certain advantages over other graph reduction techniques for training GNNs.

**Strengths:**

1. The scalability issue of GNNs is an important challenge.
2. I think this is the first work to study coresets for GNNs.

**Weaknesses:**

1. I think the formulation of the coreset problem proposed in the paper is not practical. Basically, it only aims to reduce the size of the training set, but at training time, the ego-graphs of them still need to be loaded, which could still be huge because of the neighbor explosion phenomenon. This is also the reason why plain mini-batch training is not applicable in the first place. The coreset problem proposed in this paper doesn't solve the memory usage challenge faced by mini-batch training techniques.

2. That being said, in the experiments, the authors should report the total size of ego-graphs. In this paper the percentage of labeled nodes  (among the entire training set) is considered as the key compression ratio, but this does not really reflect the memory usage in the case when mini-batch training is already inapplicable.

3. The motivation of the proposed method is to scale up GNN training; however, from the empirically results, the proposed method has similar performance as uniform sampling on two largest data sets, namely arxiv and products. Moreover, the accuracy on these two datasets and Reddit are much lower than full graph training, e.g., 53 vs. 75 on products. From the official leader board, even MLP can achieve 61.

4. The proposed method relies on a strong assumption (assumption 1), which is verified only on Cora a very small graph. I think this might not be the case on large graphs, which are the main target of the proposed methods. This could also be the reason why the proposed method has poor performance on relatively large datasets.

**Questions:**

Does assumption 1 holds on arxiv and products?

---

### Official Review · Reviewer_76UD · 2023-11-01

**Soundness:** 4 excellent
**Presentation:** 4 excellent
**Contribution:** 3 good
**Rating:** 8
**Confidence:** 4

**Summary:**

In this paper, the authors aim to solve the computational efficiency problem of GNN by proposing a coreset selection algorithm for GNN in node classification tasks. The proposed method named spectral greedy graph coresets(SGGC) is a two steps greedy algorithm: 1) it first find a large set of ego-graphs that are distance from each other to cover the topology information of graph as much as possible, 2) then it filter out subgraphs that are not contributing to optimization of the classification loss. This “extrapolate” and “exploit” approach effective compress the size of training data without pre-training while maintaining the downstream GNN performance in node classification.  The authors provided both theoretical guarantee and extensive experimental validation on multiple large datasets. The method works well with different GNN structure and doesn’t require any pre-training or repetitive inference, which improves it efficiency and generalization capability.

**Strengths:**

Pro:
- Very well written and easy to follow each step, with intuitive explanation and detailed proof defer to appendix. Adding alias to equation and definitions make it much easier to follow the context without tracing back to equation numbers.
- Overall very interesting idea and well executed. Decoupling the sampling step and loss optimization makes a lot of sense and well justified by both theoretical guarantee and extensive experimental results.
- Step by step decomposition of final objective and flow chart make it very intuitive to understand the theoretical analysis. Even though I didn’t go through the detailed proof in appendix line by line, I am fairly confident it’s doable.
- The decoupling nicely combined different lines of works from spectral embedding coverage and spectral linear classifiers, with better efficiency compared to directly using simple coreset selection part from CRAIG.
- Detailed coverage of existing works and their limits in node classification tasks.
- Very comprehensive experimental section that covers different real world use cases and showcase the efficacy and efficiency of SGGC. Ablation study further validate the combination of NAC and LCC is crucial and provide best results when used together.

**Weaknesses:**

Cons:
- Small typos in second paragraph of “Graph spectral domain” it should be “1 \geq 1 - 1/2 \lambda_1 \geq  … \geq 1 - 1/2 \lambda_n \geq 0. “
- The notation of $\tidle{Z}_i$ is a bit confusing. It is using the diffusion ego-graph $\tidle(G)_i$ for A and X, but has $G_i$ in them. Maybe I am miss understanding the notation here but it’s hard to connect it back with p for the RSD analysis.
- Figures y-axis show accuracy value between 0 and 1, yet y label says %. Is this also a typo or the accuracy is 0.8%? It would be better to use different line type besides color coding as well.

**Questions:**

- Why is Flickr doing much better with 2% ratio in SGGC, even compare to oracle full graph? Are the underlying GNN well-trained?
- The performance gap in larger dataset do seem like more significant when compare to mid/smaller dataset, any explanation or potential improvements?
- Table 2 and 3 have numbers outside of best range but still in bold? What are the indication of these numbers?

---

### Official Review · Reviewer_CXBD · 2023-11-01

**Soundness:** 3 good
**Presentation:** 3 good
**Contribution:** 3 good
**Rating:** 6
**Confidence:** 4

**Summary:**

This paper proposes a novel graph coreset method, called Spectral Greedy Graph Coresets (SGGC), to accelerate the training of graph neural networks (GNNs) by using less training data. SGGC adopts a two-stage training strategy: It first coarsely selects the widely spread ego graphs and then refines the selections by considering the diversity of its topology. A greedy algorithm is proposed to approximate these two training objectives. Extensive experiments validate the effectiveness of the proposed method over various graph reduction methods, including sampling, coarsening, and condensation.

**Strengths:**

1. This paper provides a new perspective on accelerating the training of GNNs, i.e., the coreset selection. Simply transferring the traditional
coreset algorithms into graph data is not an optimal solution. This paper overcomes the complexity dependence between nodes and proposes an effective method.

2. By leveraging the spectral embedding of nodes, which represents the node positions in a graph, SGGC can select the coreset ego-graphs with effective and diverse structures and preserve the crucial structural information.

3. This paper theoretically proves that the proposed algorithm can approximately solve the graph corsets problem, resulting in a good trade-off between effectiveness and efficiency.

4. Extensive experiments convinced me of the effectiveness of the proposed method.

**Weaknesses:**

1. What is the complexity of the proposed method? Does it comparable to other graph reduction methods, e.g., coarsening and condensation.

2. Does the proposed method suitable for inductive setting? For example, in the Cora dataset, some training nodes exists in a small connected component. In this situation, does the spectral embedding still work?

**Questions:**

See weaknesses.

---

### Meta-Review · Area_Chair_iF6V · 2023-12-05

**Metareview:**

The paper introduces a new method for speeding up the training of GNN. The main idea behind the paper is to build a coreset by identifying important subgraphs via spectral embeddings.

The paper contains some interesting ideas but some important limitations have been highlighted during the review process:

- the experimental results on large graphs are not too convincing

- the computation complexity of the method is a bit problematic

- the theoretical analysis relies on untested assumptions

Overall, the paper has some merits but it is below the acceptance bar of ICLR.

**Justification For Why Not Higher Score:**

- the experimental results on large graphs are not too convincing

- the computation complexity of the method is a bit problematic

- the theoretical analysis relies on untested assumptions

**Justification For Why Not Lower Score:**

N / A

---

### Decision · Program_Chairs · 2024-01-16

Reject